# Hyperbolic polaritonic crystals with configurable low-symmetry Bloch modes

Jiangtao Lv[1,2,15], Yingjie Wu [3,15] ✉, Jingying Liu[4,5,15], Youning Gong[6], Guangyuan Si[7], Guangwei Hu [8], Qing Zhang [9], Yupeng Zhang[6], Jian-Xin Tang [4,10], Michael S. Fuhrer [11,12], Hongsheng Chen [3], Stefan A. Maier [11,12,13], Cheng-Wei Qiu [14] ✉ & Qingdong Ou [4,5,11] ✉

Photonic crystals (PhCs) are a kind of artificial structures that can mold the flow of light at will. Polaritonic crystals (PoCs) made from polaritonic media offer a promising route to controlling nano-light at the subwavelength scale. Conventional bulk PhCs and recent van der Waals PoCs mainly show highly symmetric excitation of Bloch modes that closely rely on lattice orders. Here, we experimentally demonstrate a type of hyperbolic PoCs with configurable and low-symmetry deep-subwavelength Bloch modes that are robust against lattice rearrangement in certain directions. This is achieved by periodically perforating a natural crystal $\alpha$-MoO$_3$ that hosts in-plane hyperbolic phonon polaritons. The mode excitation and symmetry are controlled by the momentum matching between reciprocal lattice vectors and hyperbolic dispersions. We show that the Bloch modes and Bragg resonances of hyperbolic PoCs can be tuned through lattice scales and orientations while exhibiting robust properties immune to lattice rearrangement in the hyperbolic forbidden directions. Our findings provide insights into the physics of hyperbolic PoCs and expand the categories of PhCs, with potential applications in waveguiding, energy transfer, biosensing and quantum nano-optics.

The manipulation of light at a deeply subdiffractional scale plays a crucial role in imaging and focusing, optical communication, integrated optical circuits, and molecular sensing[1–5]. Polaritons, hybrid quasi-particles originating from the coupling of photons and material excitations, open up a promising pathway to light control[6–11], owing to their subwavelength scales and strong field compression capability. Particularly, ultra-confined low-loss polaritons in van der Waals materials have been hailed for next-generation nano-optical device applications; thus, various engineering strategies have been developed, including heterostructures[12–19], resonators and metasurfaces[20–23], twist-optics[24–26], and structured polaritonics[27,28].

Polaritonic crystals (PoCs), with an analogous formalism to the well-known photonic crystals (PhCs)[29], have been demonstrated to be an effective approach to molding the flow of polaritons[30–35]. The periodic variation of permittivity in PoCs can introduce the Bragg

scattering of propagating polaritons. Many polariton modes have thus been achieved, allowing for appealing potential applications, such as waveguiding and splitting[36]. Prior works for PoCs were mainly based on polaritons with in-plane isotropic wavevectors ($k_x = k_y$), for example, phonon polaritons (PhPs) in hBN[31,34,35] or plasmon polaritons in graphene[32,33]. The Bloch modes in those PoCs are highly symmetric and sensitive to lattice periodicity, similar to conventional isotropic PhCs. Recently, anisotropic spectral responses have been reported in arrays made of in-plane anisotropic $\alpha$-MoO$_3$ slabs ($k_x \neq k_y$)[37–39], but relevant Bloch modes and symmetry properties have not been studied, which hinders the unlocking of the full potential of such anisotropy in PoCs.

In this work, we design and fabricate a type of hyperbolic PoCs made of perforated $\alpha$-MoO$_3$ slabs with nanometric thicknesses. The hyperbolic PoCs, operating in the mid-infrared range, exhibit low-symmetry Bloch modes that can be tuned by the twist angle and

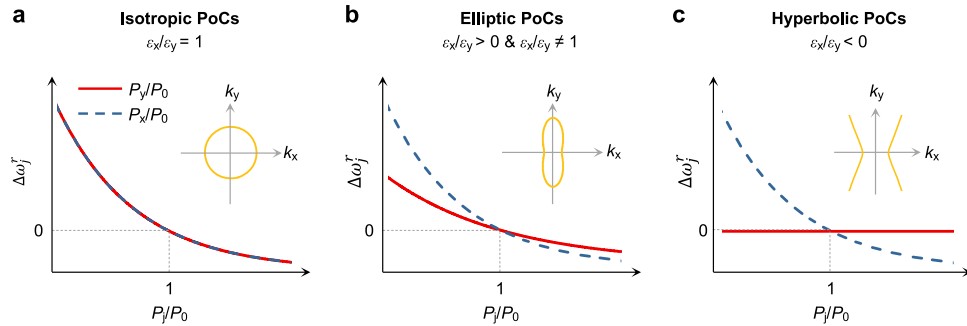

**Fig. 1 | Scaling law of polaritonic crystals (PoCs). a** Resonance shifts ($\triangle\omega_j^r$) of isotropic PoCs as a function of lattice periodicity. $P_j$ ($j = x, y$) is the periodicity along the $x$ and $y$ directions, respectively. $P_0$ is a constant period. $\varepsilon_x$ and $\varepsilon_y$ are dielectric permittivities along the $x$ and $y$ crystalline axes. Insets show the isofrequency contours of the media, where $k_x$ and $k_y$ represent polariton wavevectors in the $x$ and $y$ directions, respectively. **b** Resonance shifts of elliptic PoCs, showing slight asymmetry when the periodicity varies. **c** Resonance shifts of hyperbolic PoCs with highly asymmetric scaling relations. The resonance peak remains unchanged with the varied periodicity in the $y$ direction.

periodicity of the lattice structures, which is fundamentally dictated by the momentum matching between reciprocal lattice vectors and hyperbolic light dispersions. More importantly, such hyperbolic PoCs have robust resonant properties immune to lattice rearrangements in the forbidden direction of hyperbolic PhPs. These results unveil the unique Bloch modes in hyperbolic PoCs and pave the way to controlling these low-symmetry and configurable modes, offering possibilities for light manipulation that are hardly attainable in conventional photonic crystals and other artificial structures.

## Results
### Scaling laws of polaritonic crystals
Based on the permittivity of the media, we now categorize PoCs under three headings: isotropic PoCs ($\varepsilon_x/\varepsilon_y = 1$), elliptic PoCs ($\varepsilon_x/\varepsilon_y > 0$, $\varepsilon_x/\varepsilon_y \neq 1$), and hyperbolic PoCs ($\varepsilon_x/\varepsilon_y < 0$). One of the key virtues of PoCs is efficient light field control by adjusting structural configurations. Using rectangular-type PoCs with a fixed defect geometry and size as an example, their resonance frequencies ($\omega_j^r, j = x, y$) are closely related to periodicity ($P_j$). Based on the Rayleigh-Wood anomaly[40], the frequency shift of isotropic PoCs, denoted by $\triangle\omega_j^r = \omega_j^r - \omega_0^r$, where $\omega_0^r$ is the polariton resonance frequency of the square lattice with $P_x = P_y = P_0$, decreases reciprocally with $P_j$ along both $x$ and $y$ directions (Fig. 1a, see Supplementary Note 1 for details), resulting from the in-plane isotropic wavevectors (Fig. 1a, inset). Conventional PoCs made from metals, and recently developed graphene and hBN all belong to this kind of PoCs. As shown in Fig. 1b, in elliptic PoCs, $\triangle\omega_j^r$ also decreases with $P_j$ but exhibits different descending tendencies along the $x$ and $y$ directions. Further enhancement of anisotropy leads to hyperbolic PoCs ($\varepsilon_x/\varepsilon_y < 0$, Fig. 1c), in which light fields are mainly concentrated within two hyperbolic sectors, and a forbidden area with almost no energy propagation forms. Such forbidden area makes $\triangle\omega_j^r$ robust against periodicity variation at certain direction (here, the $y$-direction). This unique property is hardly attainable in isotropic and elliptic PoCs and might be useful in disorder-tolerant optical resonators, directional light beaming, and other appealing applications.

### Anisotropic Bloch modes
The hyperbolic PoCs in our study are composed of a periodically perforated α-MoO$_3$ slab on a SiO$_2$ substrate, as shown schematically in Fig. 2a. A set of $5 \times 5$ hole arrays with a fixed diameter ($d = 0.6$ μm) yet varied periodicities ($P$, also known as lattice constant), rotation angles ($\theta$), and lattice structures were fabricated using a focused ion beam etching technique (Methods). The thickness ($t$) of the α-MoO$_3$ slab is 235 nm according to atomic force microscopy measurement (Supplementary Fig. 2). The scanning electron microscope (SEM) image of a square-lattice PoC is displayed in Fig. 2b. More SEM results can be found in Supplementary Fig. 2. The low defect of our α-MoO$_3$ crystals

leads to a long propagation length of polaritons ($L = 6.7 \pm 0.8$ μm, Supplementary Fig. 3), fulfilling the prerequisite of PoCs ($L > P$). The background colourmap in Fig. 2c relates to the amplitude of the Fast Fourier Transform (FFT) of this PoC, in accordance with the reciprocal space points (grey circles). The amplitudes of the FFT exhibit square-type distribution and decrease rapidly in higher orders.

Due to its unique crystal structures, α-MoO$_3$ supports anisotropic PhPs between its transverse (TO) and longitudinal (LO) optical phonon frequencies[9,41]. Here we focus on the frequency band ranges from 824 to 974 cm$^{-1}$, where only the permittivity along the $x$ direction (that is, the [100] crystalline axis) is negative (Fig. 2d). PhPs with in-plane hyperbolic dispersion are hosted, as confirmed by the electric field distribution, Re($E_z$), excited by a $z$-polarized point dipole (Fig. 2d, inset). A hole in α-MoO$_3$ can also launch PhPs[24]. When $d \ll \lambda_p$, where $\lambda_p$ is the polariton wavelength, the hole can be treated approximately as a dimensionless point dipole. In our study, however, $d$ is comparable to $\lambda_p$ and the geometry of the hole is non-negligible. As shown in the inset of Fig. 2e, apart from hyperbolic wavefronts, a focusing effect emerges at both left and right sides of the hole, which is caused by its convex rim, similar to the focusing enabled by a gold disc antenna[42]. In two-dimensional arrays, the hole-generated PhPs further interact with each other and excite collective modes[43]. However, the limited area of our hole arrays make it difficult to study the resonance behaviours from the far field, for example, Fourier transform infrared (FTIR) spectra, due to the relatively weak resonance strength of collective modes (Supplementary Fig. 4). We thus calculated its absorption spectrum numerically and plotted in Fig. 2e (blue curve). A peak at 823 cm$^{-1}$ can be found, corresponding to the TO of α-MoO$_3$ with a slight red shift caused by the SiO$_2$ substrate. Besides, more resonant peaks at higher frequencies appear, due to the scattering of wavefronts at holes and subsequent Bragg resonances[38].

To analyze the interaction between propagating PhPs and periodic holes in hyperbolic PoCs, we plot the isofrequency contours (IFCs) of PhPs at the frequencies of absorption peaks in Fig. 2c (hyperbolic curves). According to the intersections of IFCs and reciprocal lattice vectors, we attribute the absorption peaks at 840, 892 and 925 cm$^{-1}$ to the (0, 0), ($\pm 1$, 0) and ($\pm 2$, 0) order Bragg resonances, respectively. Such assignment is further confirmed experimentally by performing FFT on near-field interference patterns (see below).

The in-plane hyperbolic wavefronts of PhPs make the resonance in hyperbolic PoCs highly directional. As seen from the highly directional Bloch mode in Fig. 2f, the resonant field is dominantly concentrated at the $x$ direction, while almost absent along the $y$ direction. This anisotropic field distribution is remarkably different from those of isotropic PoCs made from in-plane isotropic polaritons[31,34] and bears potential applications in polariton guiding, routing, and collimation. We then analyze its band structure by scanning along the edges of the first

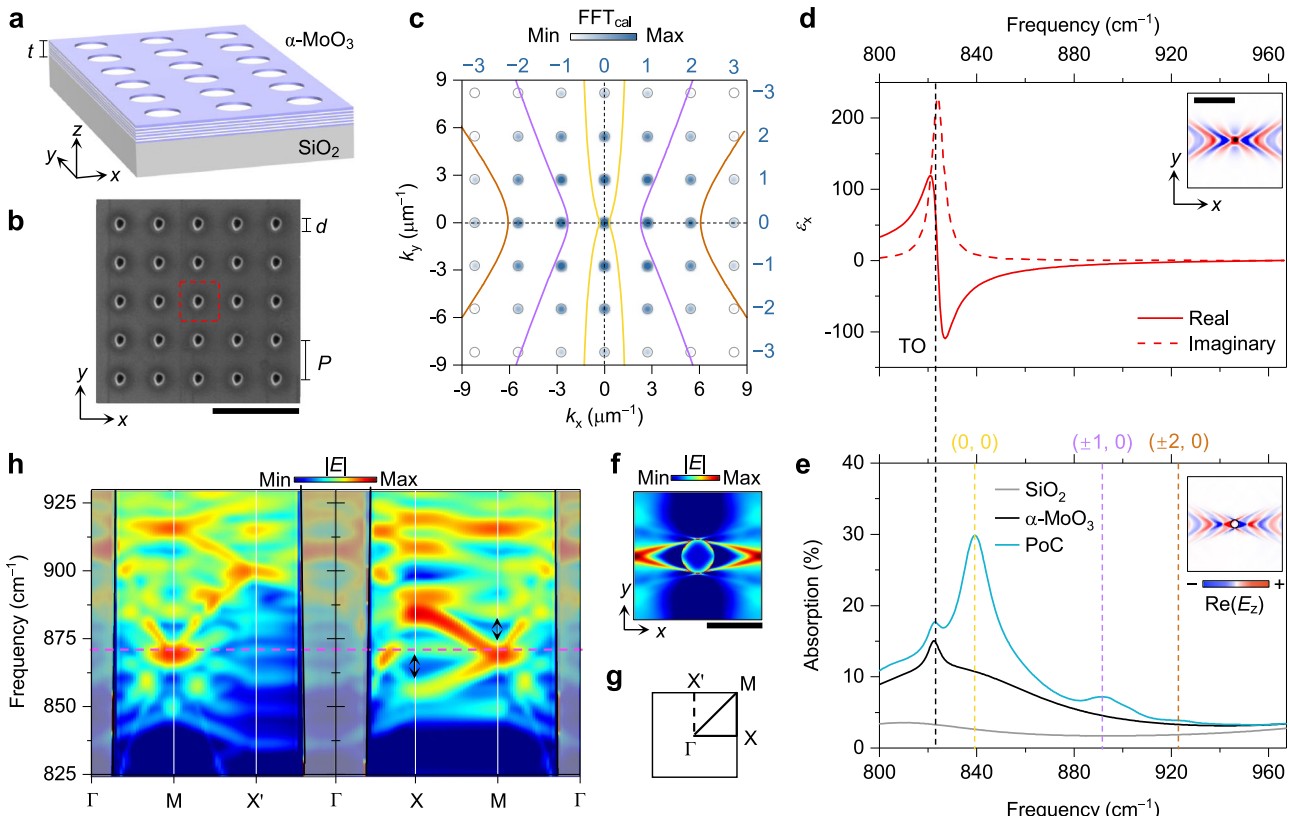

**Fig. 2 | Hyperbolic polaritonic crystals (PoCs) made of a periodically perforated α-MoO₃ slab. a** Schematic of a square-lattice PoC with the thickness ($t$) of 235 nm. **b** Scanning electron microscope (SEM) image of a square-lattice PoC with the periodicity ($P$) of 2.3 μm and diameter ($d$) of 0.6 μm. Red dashed square indicates the unit cell. Scale bar, 5 μm. **c** Calculated isofrequency contours (IFCs) of phonon polaritons (PhPs) and Fast Fourier Transform (FFT$_{cal}$) amplitude maps of the PoC. Grey circles represent reciprocal space points. The yellow, purple, and brown curves were calculated at 840, 892, and 925 cm⁻¹, respectively. **d** Optical permittivity ($\varepsilon_x$) of α-MoO₃ along the $x$ direction. Inset, simulated hyperbolic wavefronts of PhPs in the α-MoO₃ slab excited by a point dipole (black dot). Scale bar, 3 μm.

**e** Calculated absorption coefficients of the SiO₂ substrate, the plain α-MoO₃ slab supported by SiO₂, and the PoC. Black, yellow, purple, and brown dashed lines indicate the transverse optical (TO) phonon frequency and Bragg resonance frequencies. Inset, simulated electric field distribution around a single hole (white dot) in the α-MoO₃ slab excited by a plane wave. **f** Simulated electric field distribution (|E|) of the Bloch mode in a unit cell at the frequency of 872 cm⁻¹ (marked by the magenta dashed line in (**h**)). Scale bar, 1 μm. **g** The first Brillouin zone of the square-type PoC. **h** Calculated band structure of the PoC. Black arrows indicate the partially opened band gaps. Grey shaded areas represent the light cones.

Brillouin zone of the hyperbolic PoC in Fig. 2g (See Methods for details). The scattering of PhPs in PoCs opens partially the photonic band gap of polaritons, as seen from the band structure in Fig. 2h and Supplementary Fig. 5. Notably, although the hyperbolic PoC has a square-lattice structure, its band structure is different when sweeping along the Γ–X–M–Γ and Γ–X′–M–Γ routes, caused by the in-plane hyperbolic wavevectors of PhPs. The quasi-flat band around 872 cm⁻¹ demonstrates the self-collimation effect in the PoC[44], because of its hyperbolic IFCs.

**Frequency and scale tunability**
Due to the resonant nature of hyperbolic PoCs, their optical properties can be configured and tuned by frequency and lattice periodicity. Three square-lattice PoCs with $P$ = 1.3, 1.8, and 2.3 μm were prepared and observed through a scattering-type scanning near-field optical microscopy (s-SNOM) at the frequencies close to resonance peaks to visualize the real-space interference patterns in PoCs (Methods). As shown in Fig. 3a, for a PoC with a fixed periodicity ($P$ = 2.3 μm), its near-field interference patterns rely closely on frequencies, which match well with simulated electric field distributions in Fig. 3b. This phenomenon is caused by the frequency-dependent wavevectors and wavelengths of PhPs. Similar effect has been observed in isotropic PoCs[31–35], but the interference patterns and field distributions are highly anisotropic in our hyperbolic PoCs. The FFT amplitude maps obtained from the near-field images in Fig. 3a are displayed in Fig. 3c, where stronger contrast

can be observed at the intersections of IFCs and reciprocal space points. This phenomenon verifies our mode assignment.

At a given frequency, the interference patterns also vary with periodicities, as seen from the near-field images and simulated field distributions of the PoCs with $P$ = 1.3 and 1.8 μm at 904 cm⁻¹ (Supplementary Fig. 6a, b). More results at their resonance frequencies are displayed in Fig. 4a–d. All those images exhibit high anisotropy. As a comparison, the near-field image at 987 cm⁻¹ is provided in Supplementary Fig. 6c with a significantly different and reduced anisotropic interference pattern due to the in-plane elliptic dispersion of PhPs. We also calculated their far-field absorption map (Fig. 4e) and corresponding spectra (Fig. 4f) of the three PoCs and found that the polariton resonance peaks shift towards higher frequencies with decreasing $P$ and form several dispersion branches (dashed curves in Fig. 4e), while the TO phonon frequency remains unchanged. We attribute this phenomenon to the smaller reciprocal lattice vectors of PoCs with larger $P$, which require the shift of IFCs to satisfy the request of Bragg resonances, namely, intersections between IFCs and FFT harmonics. This tendency can be verified by the stronger contrasts of FFT harmonics that interact with IFCs in Fig. 4g, h, indicating the resonance orders of (0, ±1) and (±1, 0), respectively.

Besides peak shift, the absorption strength also varies with $P$ and reaches the maximum at the optimal resonance condition, that is, IFCs intersecting most reciprocal space points with high amplitude. As seen from Fig. 4f, for the PoC with $P$ = 1.3 μm, the highest absorption peak is

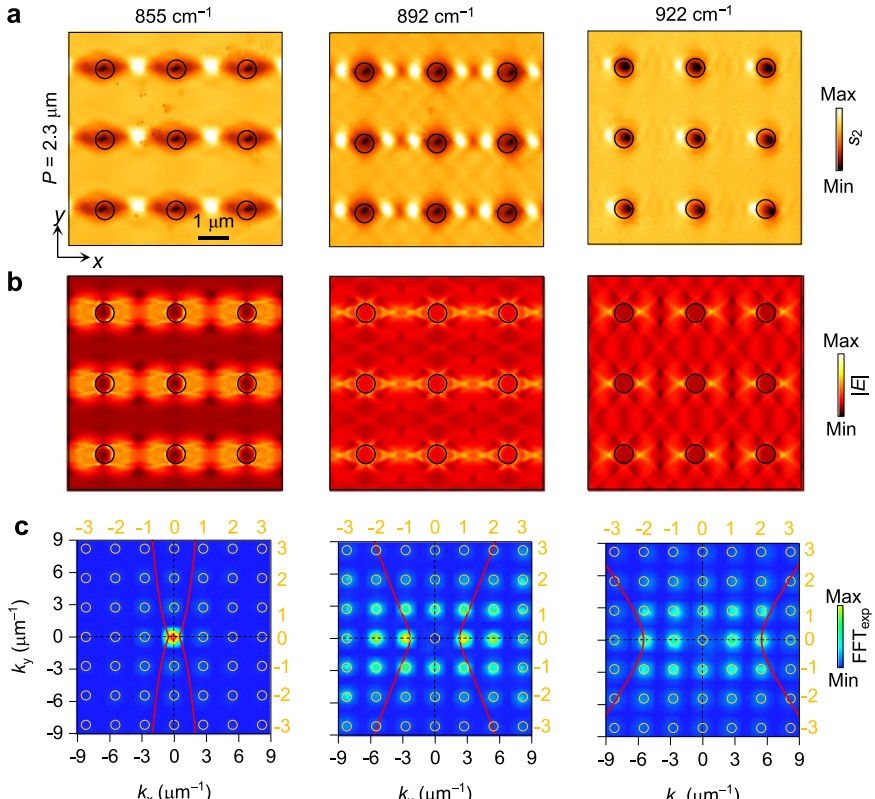

**Fig. 3 | Frequency-dependent collective modes in hyperbolic PoCs.**
**a** Experimentally measured near-field amplitude images of PoCs ($P = 2.3\,\mu m$) at frequencies close to polariton resonance frequencies. **b** Numerically simulated electric field distribution images (normalized to the one at 855 cm$^{-1}$). Scale bar for (**a**) and (**b**), 1 μm. **c** IFC contours (red curves) and Fast Fourier Transform (FFT$_{exp}$) amplitude maps of the measured near-field images in (**a**). The mode orders were determined by the contrast of FFT maps as well as the intersections between IFC contours and reciprocal space points (orange circles).

located at 871 cm$^{-1}$, corresponding to the (0, ±1) resonances (Supplementary Fig. 7a). This resonance peak merges gradually with the (0, 0) resonance peak as $P$ increases and reaches the maximum when $P = 1.5\,\mu m$ at around 861 cm$^{-1}$ (Supplementary Fig. 7b). Once $P$ further increases, the intensities of FFT amplitudes decrease fast, leading to weaker resonances. The resonance of PoCs can also be tuned by controlling the diameter of holes, as seen from calculated absorption spectra in Supplementary Fig. 8a. However, the diameter mainly relates to the amplitude intensity of FFT maps, imposing no influence on the reciprocal space points and thus leading to smaller frequency shifts compared to that of periodicity for the PoCs with a fixed $d/P$ ratio (Supplementary Fig. 8b). Besides, we also calculated the band structures of PoCs with varied scales (Supplementary Fig. 9). The lattice periodicity also plays a major role in the band structure of PoCs. It can shift the band frequencies of PoCs and change relative field intensities. The diameter, however, has a limited impact on band structures.

## Low symmetry via rotation control

The natural in-plane hyperbolic PhPs enable rotation-tunable low-symmetry Bloch modes in our PoCs, offering an effective degree of freedom for resonance control, which is completely distinct from isotropic PoCs made of graphene or hBN. We prepared a series of square-lattice PoCs with their arrangement directions deviating from the crystalline axis of α-MoO$_3$ (Methods), denoted by the angle ($\theta$) between the PoC edge and the $x$ crystalline axis. The near-field amplitude images of PoCs with $\theta = 15$, 30, and 45° at their resonance frequencies are shown in Fig. 5a–c. Because the interferences and resonances of PoCs are symmetric at $\theta = 45°$, the behaviours of PoCs at the $\theta$ from 45° to 90° are not shown here. The interference pattern changes with the increase of $\theta$ due to the change of the relative position between holes, while the field distribution of a single hole is barely

influenced, which agrees well with the full-wave simulations (Fig. 5d–f). The Bragg resonances enable dynamic tuning of the absorption spectra during the rotation of hyperbolic PoCs, as seen from Fig. 5g. We plot the IFC curves in Fig. 5i–k at the resonance frequencies marked by coloured dots in Fig. 5h. The rotation of arrays changes reciprocal lattice vectors (FFT maps in Fig. 5i–k), while IFCs remain unchanged, leading to shifted resonance frequencies similar to those of PoCs with different periodicities. However, the mode assignment and superposition of resonances are relatively complicated in rotated hyperbolic PoCs, because of their oblique reciprocal lattice vectors.

## Robustness against lattice rearrangement

We now analyze the polaritonic behaviours of different lattice structures. A diamond-type hyperbolic PoC with $P = 2.3\,\mu m$ was fabricated. This PoC has a similar arrangement to the square-type PoC rotated by 45°, however, the distance between the two nearest holes is always $P$ in the former. The near-field image at 892 cm$^{-1}$ is displayed in Fig. 6a, where the destructive interference is observed between the holes, consistent with the simulated field distribution (Fig. 6b). From the absorption spectra in Fig. 6c, surprisingly, we find that the strongest absorption peaks corresponding to the (0, 0) resonances of the diamond- and square-type PoCs lie at almost the same frequency. We attribute this nontrivial phenomenon to the in-plane hyperbolic dispersion of PhPs. Out of the two hyperbolic sectors, the propagation of polaritons is forbidden. The change of lattice vectors in this forbidden area (grey shaded areas in Fig. 6d, e) has limited influence on the absorption of PoCs (Fig. 1c). To verify this hypothesis, we investigate theoretically a set of rectangle-type PoCs with a fixed $x$-periodicity ($P_x = 2.3\,\mu m$) but varied $y$-periodicity ($P_y$). Their absorption spectra in Fig. 6f shows that $P_y$ plays a minor role in the resonance frequency once $P_x$ is fixed, especially when $P_y/P_x > 1$, although the strength of the

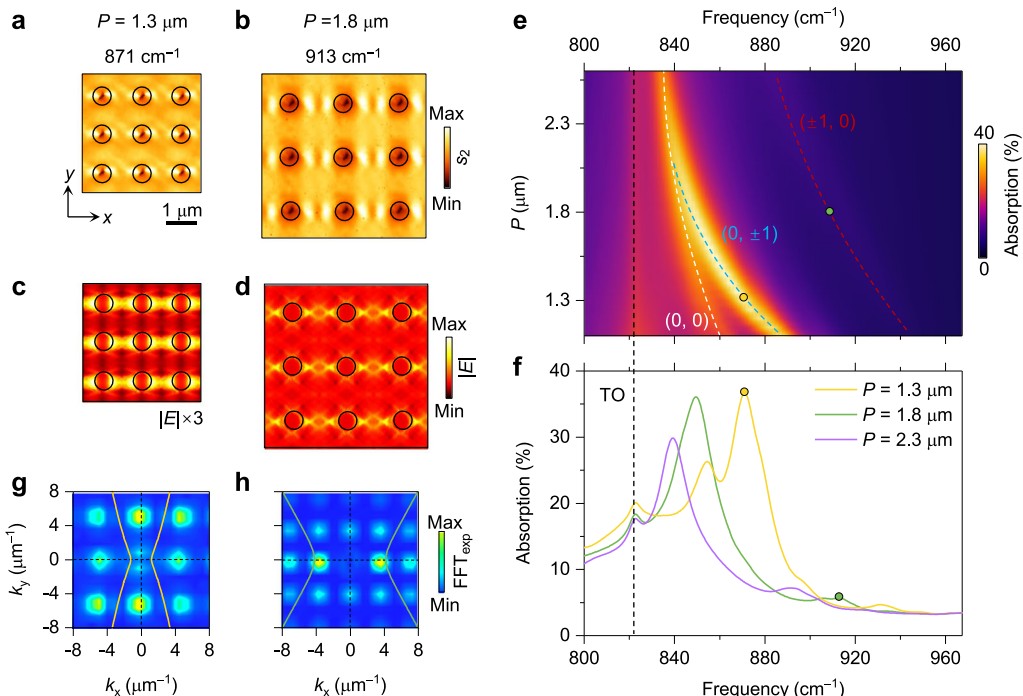

**Fig. 4 | Tuning PoC modes and resonances by varying lattice periodicity.**
**a**, **b** Near-field amplitude images of PoCs with different periodicities.
**c**, **d** Corresponding simulated electric field distributions (normalized to the one at 913 cm⁻¹). Scale bar for (**a**–**d**), 1 μm. **e** Calculated absorption coefficients of PoCs as a function of frequency and periodicity. White, blue, and red dashed curves represent the (0, 0), (0, ±1), and (±1, 0) order branches. Black dashed curve indicates the pristine transverse optical (TO) phonon mode. **f** Absorption spectra of PoCs. The yellow and green dots indicate the frequencies in (**a**) and (**b**). **g**, **h** Corresponding FFT maps of the experimental near-field images in (**a**) and (**b**). The yellow and green curves represent the calculated IFCs of PhPs at 871 and 913 cm⁻¹, respectively.

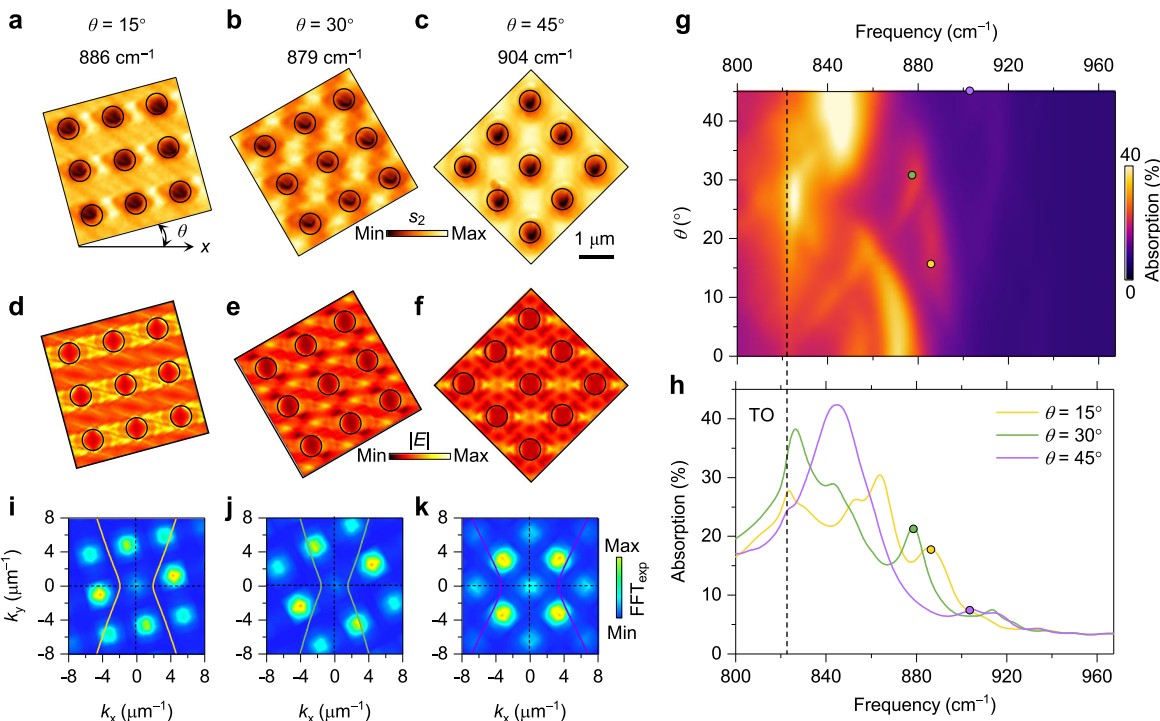

**Fig. 5 | Low-symmetry Bloch modes in rotated hyperbolic PoCs. a–c** Resonant interference patterns of PoCs with the same periodicity ($P = 1.3$ μm) but different rotation angles. **d–f** Simulated electric field distributions (normalized to the one at 886 cm⁻¹). Scale bar for (**a**–**f**), 1 μm. **g** Calculated absorption coefficients of PoCs as a function of frequency and rotation angle. **h** Absorption spectra of PoCs. The coloured dots mark the frequencies in (**a**–**c**). **i–k** Corresponding FFT maps of the experimental interference patterns in (**a**–**c**). The yellow, green, and purple curves represent the calculated IFCs of PhPs at 886, 879, and 904 cm⁻¹, respectively.

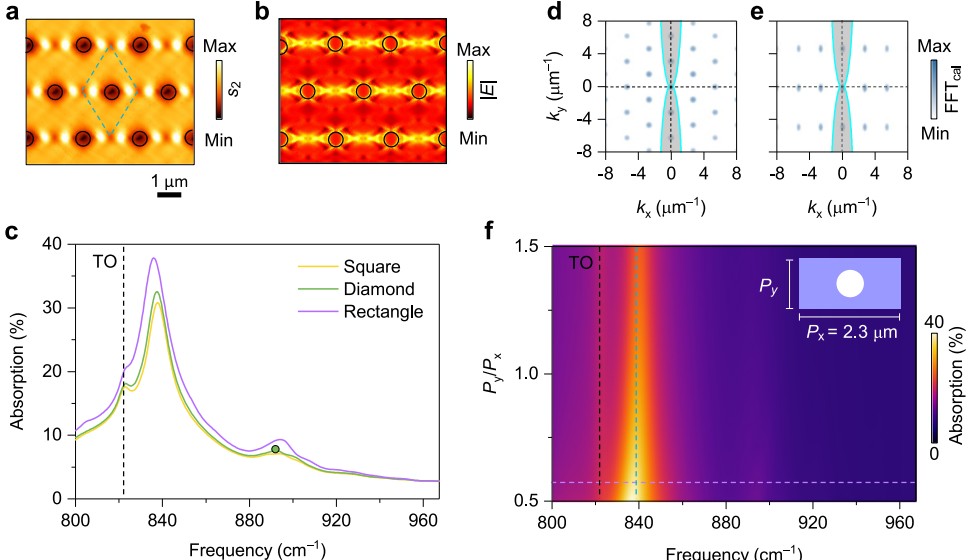

**Fig. 6 | Robust resonant modes against lattice rearrangement in hyperbolic PoCs. a** Near-field amplitude image of a diamond-lattice PoC with $P = 2.3\ \mu m$ at $892\ cm^{-1}$. Green dashed rhombus indicates the unit cell. **b** Corresponding electric field distribution. Scale bar for (**a**) and (**b**): 1 μm. **c** Absorption spectra of PoCs with different lattice arrangements. The green dot represents the frequency in (**a**). **d**, **e** IFCs of PhPs at $839\ cm^{-1}$. The colour plots represent the normalized FFT amplitude maps of PoCs with diamond (**d**) and rectangle (**e**) lattices. The grey shaded areas indicate the forbidden areas of hyperbolic dispersion. **f** Calculated absorption coefficients of PoCs as a function of frequency and length-width ratio $(P_y/P_x)$. Inset is the schematic of the rectangular-type PoC. Purple dashed line at $P_y/P_x = 0.57$ marks the absorption curve (purple) shown in (**c**). Black and cyan dashed lines indicate the frequencies of the TO phonon resonance and (0, 0) polariton resonance, respectively.

absorption slightly increases with the decrease of $P_y$. This lattice-rearrangement robustness in the hyperbolic forbidden direction makes hyperbolic PoCs a versatile building block for potential disorder-immunity applications. Note that the polariton resonance is intrinsically bound to the $x$ crystallin axis in the considered frequency ranges, yielding quasi-1D field distributions in our 2D-arranged PoCs. The robustness emerges when the lattice arrangement direction is normal to the quasi-1D field distributions, namely, the $x$ crystalline axis ($\theta = 0°$), and the robustness gradually vanishes in tilted PoCs with increasing $\theta$, as confirmed by the simulated absorption coefficient maps of rotated PoCs with varied periodicities in Supplementary Fig. 10.

## Discussion

In this work, we have designed, fabricated, and investigated hyperbolic PoCs made from in-plane anisotropic α-MoO₃ slabs. The configurable Bloch modes of hyperbolic PhPs in our PoCs are clarified by full-wave numerical simulations and real-space nano-imaging. The excitations of low-symmetry Bloch modes and Bragg resonances in hyperbolic PoCs can be engineered by controlling frequency, lattice scale, and rotation. We also find that the resonances of hyperbolic PoCs have a robustness against lattice rearrangement at certain directions out of hyperbolic sectors, which is distinct from isotropic and elliptic PoCs.

Hyperbolic PoCs add a member to the catalogue of PhCs. Endowed with in-plane anisotropy, such two-dimensional periodic structures could lead to more interesting phenomena and richer physics by designer geometries, such as triangles[28]. In combination with external stimuli, such as electric gating, we envision active tuning of the properties of hyperbolic PoCs. Our findings may thus establish a platform for optical resonators, infrared detection, enhanced molecular sensing, to name a few.

## Methods

### Sample preparation

Thin α-MoO₃ slabs were prepared by mechanically exfoliating bulk crystals[45]. PoCs were fabricated in a large-area α-MoO₃ slab using a focused ion beam microscope (FEI Helios Nanolab 600) with an electron source that is capable of capturing high-resolution scanning electron microscopy images for device quality check right after patterning. A 28 pA patterning current (gallium ions) was selected with 30 kV accelerating voltage to minimize any potential damage to the functional areas in the α-MoO₃ slab with an optimized period of total milling time. Note that the PoCs were etched through α-MoO₃ slabs precisely to minimize overmilling and the redeposition effect by calibrating a test sample first.

### Infrared nano-imaging

A commercially available s-SNOM (Neaspec) was used to conduct real-space imaging experiments, in which the basic setup is based on a tapping-mode atomic force microscope (AFM). A $p$-polarized laser beam with tunable mid-infrared wavelengths was directed to the AFM tip (NanoWorld) with the tip oscillating frequency of ~285 kHz and amplitude of ~70 nm, respectively. A nanoscale hotspot was then generated on the tip apex, which acts as a near-field probe mapping the polariton field and Bragg resonances of α-MoO₃ PoCs. The tip-scattered field was finally recorded by a pseudo-heterodyne interferometer in the far field, producing near-field images by demodulating the tip oscillation harmonics.

### Calculations and simulations

The IFCs of α-MoO₃ were calculated from the dispersion relations of waveguide-mode PhPs at the given frequencies[9,46]. The near-field interference images were simulated by the finite element method using COMSOL Multiphysics. The electric field distribution of plain α-MoO₃ shown in the inset of Fig. 2d was excited by a vertically polarized dipole source. In the simulations of PoCs, we considered a 3 × 3 hole array supported by a SiO₂ substrate. A periodic plane wave at the given frequency was used as the light source to excite PhPs. We note that in our experiments hole arrays were fabricated at the centre of an α-MoO₃ flake. The edge-launched/-reflected PhPs have limited contributions to near-field measurements. In addition, the tip-launched PhPs were significantly suppressed because the first-order FFT has no intersections with the IFC of tip-launched PhPs[28]. Considering that a

hole itself can launch PhPs effectively (Fig. 2e inset), we thus used a plane wave to excite PhPs in our simulations. The electric field at the plane 10 nm above the α-MoO₃ flake surface was recorded. The absorption spectra of the system were calculated by the same method but performed on a unit cell. The band structure of PoCs was calculated numerically by the finite difference time domain (FDTD) method using Ansys Lumerical. In the simulation, a unit cell was placed on a SiO₂ substrate. Bloch boundary conditions were set up at the sidewalls of the unit cell, whereas the top and bottom surfaces were set as perfectly matched layers. Randomly polarized dipoles were used as the light source to excite different Bloch modes[31,34]. The signals were recorded by randomly placed monitors. Because of the relatively high losses in the system, the simulation time should be long enough. In our theoretical analysis and simulations, the thickness of α-MoO₃ was set as 235 nm. The permittivity values of α-MoO₃ were referred to the previous study[46].

## Data availability

The Source Data underlying the figures of this study are available at https://doi.org/10.6084/m9.figshare.22721188. All raw data generated during the current study are available from the corresponding authors upon request.

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

## Acknowledgements

This project was supported by the Science and Technology Development Fund, Macau SAR (No. 0116/2022/A3), the Australian Research Council (DE220100154 and CE170100039), the National Natural Science Foundation of China (Grant No. 62105058) and the Fundamental Research Funds for the Central Universities Key Scientific Research Guidance Project (Grant No. N2023005). The work at Zhejiang University was sponsored by the Key Research and Development Program of the Ministry of Science and Technology (Grants Nos. 2022YFA1404704, 2022YFA1405200, and 2022YFA1404902), the National Natural Science Foundation of China (Grant No. 61975176), the Key Research and Development Program of Zhejiang Province (Grant No. 2022C01036), and the Fundamental Research Funds for the Central Universities. G.H. acknowledges the startup grant from Nanyang Technological University. S.A.M. additionally acknowledges the Lee-Lucas Chair in Physics. This work was performed in part at the Melbourne Centre for Nanofabrication (MCN) in the Victorian Node of the Australian National Fabrication Facility (ANFF).

## Author contributions

J.T.L., Y.W. and J.Y.L. contributed equally to this work. Y.W., C.-W.Q. and Q.O. conceived the idea. J.T.L., J.Y.L. and Q.O. conducted material preparation and optical measurements. Y.G. and Y.Z. contributed to optical measurements. Y.W. and H.C. performed theoretical analysis and numerical simulations. G.H., Q.Z., J.-X.T., M.S.F. and S.A.M. contributed to analysis and discussion of the results. Y.W., G.S., and Q.O. co-wrote the manuscript with inputs and comments from all authors. C.-W.Q. and Q.O. supervised the project.

## Competing interests

The authors declare no competing interests.

## Additional information

[1]College of Information Science and Engineering, Northeastern University, Shenyang 110004, China. [2]School of Control Engineering, Hebei Key Laboratory of Micro-Nano Precision Optical Sensing and Measurement Technology, Northeastern University at Qinhuangdao, Qinhuangdao 066004, China. [3]ZJU-Hangzhou Global Scientific and Technological Innovation Center, Zhejiang University, Hangzhou 311215, China. [4]Macao Institute of Materials Science and Engineering (MIMSE), Faculty of Innovation Engineering, Macau University of Science and Technology, Taipa, Macao 999078, China. [5]Department of Materials Science and Engineering, Monash University, Clayton, Victoria 3800, Australia. [6]State Key Laboratory of Radio Frequency Heterogeneous Integration, College of Electronics and Information Engineering, Shenzhen University, Shenzhen 518060, China. [7]Melbourne Centre for Nanofabrication, Victorian Node of the Australian National Fabrication Facility, Clayton 3168 VIC, Australia. [8]School of Electrical and Electronic Engineering, Nanyang Technological University, Singapore 639798, Singapore. [9]School of Physics, University of Electronic Science and Technology of China, Chengdu 611731, China. [10]Jiangsu Key Laboratory for Carbon-Based Functional Materials & Devices, Institute of Functional Nano & Soft Materials (FUNSOM), Soochow University, Jiangsu 215123, China. [11]ARC Centre of Excellence in Future Low-Energy Electronics Technologies, Monash University, Clayton, VIC 3800, Australia. [12]School of Physics and Astronomy, Monash University, Clayton, VIC 3800, Australia. [13]Department of Physics, Imperial College London, London SW7 2AZ, UK. [14]Department of Electrical and Computer Engineering, National University of Singapore, Singapore 117583, Singapore. [15]These authors contributed equally: Jiangtao Lv, Yingjie Wu, Jingying Liu. ✉e-mail: yingjie.wu@zju.edu.cn; chengwei.qiu@nus.edu.sg; qdou@must.edu.mo

