## [Peer Review File · Nature Communications]

Hyperbolic polaritonic crystals with configurable low-symmetry Bloch modesREVIEWER COMMENTS

Reviewer #1 (Remarks to the Author):

Jiangtao Lv and co-authors report the study of highly anisotropic polaritonic crystals made of periodically perforated flake of bi-anisotropic van der Waals crystal MoO₃. They employ near-field imaging and study propagating phonon-polaritons in the periodic structure, and claim observation of anisotropic Bloch modes. I find this work interesting and timely, and in principle suitable for publication in Nature Communications. However, I have major concerns regarding the main claims of this work. Furthermore, the quality of the data analysis must be significantly improved before the manuscript can be considered for publication. Below I listed my concerns and questions.

Major concerns

1. I am not convinced that the experimental results demonstrate the presence of polaritonic Bloch modes in the samples. Polaritons in the second reststrahlen band of MoO₃ are known to propagate just 2-3 wavelengths before they dissipate [see for example Appl. Phys. Lett. 120, 113101 (2022), Adv. Opt. Mater. 10, 2102057 (2022), and Adv. Opt. Mater. 10, 2201492 (2022)]. I would expect that such short propagation length would hinder formation of Bloch modes in the array, especially considering the scattering at holes edges. In the recent near-field studies of hBN-based PoC (Ref. 30,33,34), Bloch modes have been shown to have distributed nature so that the hole geometry almost disappears from the near-field images due to continuous LDOS distribution, in contrast to the local interference pattern formed around the holes edges which are clearly visible. In this work, presented near-field maps (fig. 3a-c and fig. S3) do not clearly demonstrate the evolution between the near-field interference corresponding to the "local" and the Bloch modes.

1.1 In order to show the existence of the Bloch modes, please provide near-field images at different frequencies showing the interference pattern of different nature – local and distributed across the array (such as in fig. 2 in Ref. 34, or fig. 3 in Ref. 33). Figure similar to fig. S3 should be included in the main text.

1.2 Please analyze the role of material loss in the formation of flat bands corresponding to the Bloch modes. Figure 2h indicates that Bloch modes are extremely lossy, which makes their observation very challenging. It will be helpful to see numerical simulations showing the Bloch modes in PoC with different material loss.

1.3 Authors used 5x5 hole arrays. Is this enough to excite collective modes? For example, authors of Ref. 30 used 25x25 holes array, and authors of Ref. 33 used 16x16 holes array.

2. Is the hole diameter d important for the PoC band structure? Please discuss the impact of the d/P ratio on the dispersion and far-field PoC response.

3. In fig. 2c and S4a, IFCs do not cross the array momenta (the FFT map), yet the selected frequencies correspond to the absorption peaks. Please explain this mismatch. Also, the clear absorption peak at around 925 cm⁻¹ in fig. 2e should be explained.

4. In fig. 3, the near-field mapping frequency of 903.8 cm⁻¹ does not correspond to any absorption feature of PoC (according to fig. 3g). Please show the near-field scans corresponding to the resonant modes in the structure.

5. Different PoC dispersion branches visible in fig. 3g and 3h should be explained.

5.1 Please explain why the absorption maximum for $P = 1.3 \mu\text{m}$ does not correspond to the zero-order resonance as for the other two period values.

Minor problems

1. In line 87, what does the "resonance frequency" mean? PhCs generally have multiple bands at different frequencies. Also, please clarify the meaning of "frequency shift" – shift relative to what

frequency?

2. I am not sure what is the importance of the section "Scaling laws of photonic crystals". It seems disconnected from the following discussion. If this section is meant to introduce the concept of hyperbolic phonon-polaritons and hyperbolic PhC based on hyperbolic materials, then please provide references and clearly discuss these two concepts in consecutive manner.

2.1 Particularly, the discussion in lines 90-97 is very confusing. It seems that authors mixed up two very different topics – anisotropic polaritons and photonic crystals. For example, what is the meaning of $\epsilon_{x/y}$? Is this the permittivity of the polaritonic material? Please clarify the discussion.

3. How was the expression in line 88 for $\delta\omega$ derived? This expression must be explained in details in supplementary information.

3.1 What is the importance of this expression? I don't see any use of it in the following analysis.

3.2 Please show how the polaritonic dispersion relation is accounted for in the expression.

3.3 Similar to the question 1 above, how does this expression correlate to the band structure of the PoC?

4. In line 150, please provide reference or analytical expression which supports the claim that quasi-flat band indicates the anisotropic Bloch mode.

5. In lines 171-172 of the main text, fig. S4b is said to correspond to $P = 1.6 \mu\text{m}$, but in the SI it is shown to correspond to $2.3 \mu\text{m}$.

6. Spectral absorption profile for 30 deg (green) in fig. 4h seems to disagree with the results in fig. 4g. The green dot in 4g supposed to be the peak position, but the actual peak is to the left from it.

Reviewer #2 (Remarks to the Author):

See attachment.

Reviewer #3 (Remarks to the Author):

In this manuscript, the authors experimentally explored a polaritonic crystal by patterning the α MoO₃ slabs. In the infrared region, the material shows hyperbolic phonon polaritons which are modified due to geometrical structuring such as changing the lattice constant or rotating the lattice of the pattern with respect to the crystal axes of α MoO₃. The authors show that the resonance shifts do not occur when the periodicity along the forbidden direction of the hyperbolic IFC is changed.

I have a general comment on the novelty and relevance of this work. Firstly, as stated in point 1) below, the claim of "robustness" of the polariton to disorder in one direction may be useless in practical situations. Secondly, regarding the novelty of the idea itself, it may be important to note that a theoretical proposal for the same system was published last year (this is reference 37 in the current manuscript):

Capote-Robayna, Nathaniel, Olga G. Matveeva, Valentyn S. Volkov, Pablo Alonso-González, and Alexey Y. Nikitin. "Twisted polaritonic crystals in thin van der Waals slabs." *Laser & Photonics Reviews* 16, no. 9 (2022): 2200428.

Hence the novelty of the idea itself is lessened. I leave to the editor the question of whether in the light of these considerations, this manuscript is worthy of publication in *Nature Communications*.

Additionally, I have a few technical comments which are listed below:

1) One of the strongly pitched novelty is the authors' claim of robustness (eg. "disorder tolerant optical resonator" in line 99) -- which seems very far fetched. In reality, their system only is robust to periodicity variation in one direction. Typically disorder would occur along all directions. Could the authors comment on what kind of experimental scenarios would give rise to this very specific kind of "disorder"?

2) A reference needs to be added for the exponential dependence of the frequency shift on the periodicity in line 88.

3) Figure 3 and its explanation in lines 163--169 is confusing. Figure 3a-c are all at a frequency of 903.8 cm⁻¹ according to the figure caption. Figure 3i-k are also presumably at the same frequency, since these are just spatial FFTs of Fig. 3a-c. However when the authors are explaining redshift of the (0,0) peaks in the paragraph starting at line 163, say "We attribute this phenomenon to the smaller reciprocal lattice vectors of PoCs with larger P, which require the shift of IFCs to satisfy the request of Bragg resonances, namely, intersections between IFCs and FFT harmonics. This tendency can be verified by the stronger contrasts of FFT harmonics that interact with IFCs in Figs. 3i-k". So the "FFT contrasts" that the authors use to explain the results are all for the same frequency of 903.8 cm⁻¹ and not at the location of the respective peaks.

4) The authors say in line 168, "For the PoC with P = 1.3 μm, the highest absorption peak is located at 871.3 cm⁻¹, which should be attributed to the (±1, ±2) resonances (Supplementary Fig. S4a). This resonance peak merges gradually with the (0, 0) resonance peak as P increases and reaches the maximum when P = 1.6 μm at around 892.4 cm⁻¹ (Fig. S4b)". However, in supplementary Figure S4b, the periodicity is chosen as 2.3 microns (according to Fig S4 caption) instead of the above stated 1.6 microns.

5) In line 193, the authors say: "Using the case with θ = 45° as an example (Fig. 4k), the highest absorption peak at 844.3 cm⁻¹ is the result of the combination of (0, 0), (±1, 0), and (0,±1) order resonances, which can be derived from the relatively stronger contrast of corresponding FFT harmonics". Once again, this is not clear from Figure 4.

The authors fabricate a hyperbolic polaritonic crystal out of the in-plane anisotropic crystal MoO₃. Although the realization of a hyperbolic polaritonic crystal has been demonstrated in previous works (see *Dickson, Wayne, et al. "Hyperbolic polaritonic crystals based on nanostructured nanorod metamaterials." Advanced Materials 27.39 (2015): 5974-5980*) in this present work the authors focus on experimental imaging of highly asymmetric Bloch modes in the natural hyperbolic crystal MoO₃ and tune them by varying array periodicity and array orientation.

1. In general, some of the main claims of the manuscript are not supported neither by experiment nor by theory:

-The authors claim that the Bloch modes are robust against defects and disorder without showing any experimental/theoretical proof. The authors name by “disorder” what is indeed a “lattice rearrangement”.

-The authors claim that the Bragg resonances exhibit robust properties to lattice rearrangement by showing a particular example of a lattice arrangement (diamond-like lattice shape). However, this statement does not hold true for other lattice arrangements (for example, if the lattice is rearranged such that the holes are displaced along the Y-axis).

2. The authors should provide more experimental data such as FTIR or nano-FTIR spectra of the arrays.

3. Quality of data presentation is a bit poor.

4. Some major comments to address are listed in the following:

- Figure 3 and 4:

- The authors should explain why simulated absorption spectra (panels g and h) do not match experiment at all. For instance, in Figure 3, for $P = 2.3 \mu\text{m}$ the (0,0) order resonance appears at 840 cm^{-1} and not at 903 cm^{-1} (big difference of about 60 cm^{-1}) as in the experimental image. In addition, why in panels e-f the simulated field distribution matches the experimental images while in the absorption spectra is completely off?

- In general, it is not clear how the mode order assignment is made. The authors should show mode field distributions for the different modes (0, 0), (± 1 , ± 1)... as well as the corresponding IFC contours and FFT map profiles (as in Supplementary Figure S2). In panels g, authors should relate mode orders to the mode branches.

- The electric field shown in panels d, e and f is E or E_z ? Is this field normalized?

- **Page 6:** The authors write $P = 1.6 \mu\text{m}$ while Fig S4b shows $P = 2.3 \mu\text{m}$: “*This resonance peak merges gradually with the (0, 0) resonance peak as P increases and reaches the maximum when $P = 1.6 \mu\text{m}$ at around 892.4 cm^{-1} (Fig. S4b).*”

5. Some minor comments are listed below:

- What is the fundamental difference of a 2D hole array of MoO₃ (with modes that only propagate in one direction) and a 1D hole array of h-BN (isotropic)?
- Can the authors show near-field images in other frequency ranges, i.e, showing the transition from hyperbolic to elliptic regimes?

Manuscript ID: NCOMMS-22-49996

Manuscript title: Hyperbolic polaritonic crystals with configurable low-symmetry Bloch modes

Point-by-point responses to Reviewers' Comments

We are very grateful for all the comments from the editor and all the reviewers. These comments are very important and valuable to improve the quality and readability of this paper. Revisions and responses to address your comments are presented as below.

Reviewer #1

Jiangtao Lv and co-authors report the study of highly anisotropic polaritonic crystals made of periodically perforated flake of bi-anisotropic van der Waals crystal MoO₃. They employ near-field imaging and study propagating phonon-polaritons in the periodic structure, and claim observation of anisotropic Bloch modes. I find this work interesting and timely, and in principle suitable for publication in Nature Communications. However, I have major concerns regarding the main claims of this work. Furthermore, the quality of the data analysis must be significantly improved before the manuscript can be considered for publication. Below I listed my concerns and questions.

REPLY: We highly appreciate the reviewer's positive comments on this work. In the revised manuscript, we have provided new experimental results according to reviewer's suggestions. New simulations and analysis have also been conducted to support our conclusions. Hope these amendments and updates can address the reviewer's concerns.

Major concerns

1. I am not convinced that the experimental results demonstrate the presence of polaritonic

Bloch modes in the samples. Polaritons in the second reststrahlen band of MoO₃ are known to propagate just 2-3 wavelengths before they dissipate [see for example Appl. Phys. Lett. 120, 113101 (2022), Adv. Opt. Mater. 10, 2102057 (2022), and Adv. Opt. Mater. 10, 2201492 (2022)]. I would expect that such short propagation length would hinder formation of Bloch modes in the array, especially considering the scattering at holes edges. In the recent near-field studies of hBN-based PoC (Ref. 30,33,34), Bloch modes have been shown to have distributed nature so that the hole geometry almost disappears from the near-field images due to continuous LDOS distribution, in contrast to the local interference pattern formed around the holes edges which are clearly visible. In this work, presented near-field maps (fig. 3a-c and fig. S3) do not clearly demonstrate the evolution between the near-field interference corresponding to the “local” and the Bloch modes.

REPLY: We agree with the reviewer that the formation of Bloch modes requires the propagation length of polaritons (L) to be long enough. To extract L , we conducted near-field measurement at 892 cm⁻¹ on the same flake but near the edge where hole arrays are absent. The obtained near-field image is shown in the inset of Fig. S3. We extracted the line trace along the gray dashed line and then fitted it using the equation

$$s(x) = A \frac{e^{-\frac{2x}{L}}}{\sqrt{x}} \sin \frac{4\pi(x-x_c)}{\lambda_p} + B \frac{e^{-\frac{x}{L}}}{x} \sin \frac{2\pi(x-x'_c)}{\lambda_p}$$

where A and B are the parameters for tip- and edge-launched PhPs, x is the distance from edge, x_c and x'_c are phase shifts, λ_p is polariton wavelength [*Nat. Commun.*, 11, 2646 (2020); *Adv. Mater.*, 32, 1908176 (2020)]. A propagation length of $6.7 \pm 0.8 \mu\text{m}$ is obtained, which is larger than the pitches of arrays. Such long propagation length allows for the formation of Bloch modes in our polaritonic crystals.

Figure S3. Extraction of propagation length.

ACTIONS: Figure S3 has been provided in the revised Supplementary Information. A sentence “The low defect of our α -MoO₃ crystals leads to a long propagation length of polaritons ($L = 6.7 \pm 0.8 \mu\text{m}$, Supplementary Fig. S3), fulfilling the prerequisite of PoCs ($L > P$).” has been added in Line 102.

1.1 In order to show the existence of the Bloch modes, please provide near-field images at different frequencies showing the interference pattern of different nature – local and distributed across the array (such as in fig. 2 in Ref. 34, or fig. 3 in Ref. 33). Figure similar to fig. S3 should be included in the main text.

REPLY: According to the reviewer's suggestion, in the revised manuscript, we have provided near-field images and corresponding simulation results at several frequencies close to resonant peaks. As our laser source doesn't include frequencies lower than 855 cm^{-1} , we used the result at 855 cm^{-1} as a substitution, which is close to the (0, 0) order resonance. For the same reason, the near-field result at 922 cm^{-1} have been provided in Fig. 3a, having a slight frequency deviation from polariton resonance frequencies of the (± 2 , 0) order resonances at 925 cm^{-1} , which will not weaken our main conclusions. As a comparison, we also provided in Fig. S6 the near-field results of different PoCs at the same frequency.

ACTIONS: Figure 3 has been added in the revised manuscript. Figure S6 has been added in the revised Supplementary Information. A sentence “This phenomenon is caused by the frequency-dependent wavevectors and wavelengths of PhPs. Similar effect has been observed in isotropic PoCs,^{31–35} but the interference patterns and field distributions are highly anisotropic in our hyperbolic PoCs.” has been added in Line 149. Besides, the sentence “At a given frequency, the interference patterns also vary with periodicities, as seen from the near-field images and simulated field distributions of the PoCs with $P = 1.3$ and 1.8 μm at 904 cm^{-1} (Supplementary Figs. S6a,b).” has been added in Line 153.

Figure 3. Frequency-dependent collective modes in hyperbolic PoCs. **a**, Near-field amplitude images of PoCs ($P = 2.3$ μm) at frequencies close to polariton resonance frequencies. **b**, Corresponding electric field distribution images (normalized). **c**, IFC contours (red curves) and FFT of the simulated images in **a**. The mode orders were determined by the contrast of FFT maps as well as the intersections between IFC contours and reciprocal space points (orange circles).

Figure S6. Near-field interference patterns and corresponding field distribution images of the PoC with $P = 2.3 \mu\text{m}$ at 904 cm^{-1} (a), $P = 1.8 \mu\text{m}$ at 904 cm^{-1} (b), the PoC with $P = 1.3 \mu\text{m}$ at 904 cm^{-1} (c) and 987 cm^{-1} (d).

1.2 Please analyze the role of material loss in the formation of flat bands corresponding to the Bloch modes. Figure 2h indicates that Bloch modes are extremely lossy, which makes their observation very challenging. It will be helpful to see numerical simulations showing the Bloch modes in PoC with different material loss.

REPLY: We agree with the reviewer that the Bloch modes in our case are highly lossy because of the large imaginary part of permittivities, $\text{Im}(\epsilon_j)$. In our FDTD simulations, $\alpha\text{-MoO}_3$ were modelled by “Sampled 3D data” with imported permittivities, including both the real and imaginary parts at the x , y , and z directions. At the beginning of simulation, the raw data (both the real and imaginary parts) were automatically fitted by the built-in equations in the software. It is technically difficult for us to arbitrarily change $\text{Im}(\epsilon_j)$ because the real part of the permittivity will be changed as well during fitting. To eliminate possible errors in fitting, we set the permittivity of $\alpha\text{-MoO}_3$ using the “Lorentz” model, in which the

frequency-dependent permittivity of α -MoO₃ is described analytically. The mode strength also depends closely on frequencies, as shown in Fig. S5 below. Note that, in the original manuscript, we only considered the TM mode excited by vertically polarized dipoles. To excite all the possible modes, we set randomly polarized dipoles in simulations during revision. Although still lossy, the obtained different band structures along the edges of the same Brillouin zone do support our main conclusion about the low-symmetry Bloch modes.

ACTIONS: Figure S5 has been added in the revised Supplementary Information to indicate the frequency dependence of the mode strength in α -MoO₃ PoCs.

Figure S5. Band structure of the PoC with $P = 2.3 \mu\text{m}$. The gray curves represent light lines. Yellow dashed rectangle surrounds the frequency domain considered in the main text.

1.3 Authors used 5x5 hole arrays. Is this enough to excite collective modes? For example, authors of Ref. 30 used 25x25 holes array, and authors of Ref. 33 used 16x16 holes array.

REPLY: The formation of collective polaritonic modes is determined by the interaction of evanescent fields between adjacent units (here are holes), which is closely dependent on their distances. The transition from isolated to collective modes has been intensively studied in plasmonic oligomers [*Nano Lett.*, 10, 2721 (2010)]. As the hole distance is smaller than the propagation length of polaritons in our system (please see the response to Comment 1.1), we

believe a 5×5 array is sufficient to excite these collective modes. The strengths of such collective modes do rely on the number of holes. That's why the authors of Refs. 30 and 33 used larger arrays. The mode strength in our 5×5 arrays is relatively weak, making the identification of the higher-order resonance peaks difficult from the FTIR spectrum (please see Fig. S4 below).

ACTIONS: We have conducted the FTIR measurement and provided the result in Fig. S4. A sentence “In two-dimensional arrays, the hole-generated PhPs further interact with each other and excite collective modes.⁴³ However, the limited area of our hole arrays make it difficult to study the resonance behaviours from the far field, for example, Fourier transform infrared (FTIR) spectra, due to the relatively weak resonance strength of collective modes (Supplementary Fig. S4). We thus calculated its absorption spectrum numerically and plotted in Fig. 2e (blue curve).” has been added in Line 117 to emphasize that the collective mode can indeed be supported by our arrays, but the mode strength is too weak to be detected from FTIR spectra. Ref. 43 has been added in the revised manuscript.

Figure S4. Reflectance spectrum of the PoC composed of 5×5 hole arrays with $P = 2.3 \mu\text{m}$. The signal was normalized to the unpatterned $\alpha\text{-MoO}_3$ region. The peaks at 817 and 1006 cm^{-1} indicate the TO and LO phonon resonance of $\alpha\text{-MoO}_3$, respectively, and overwhelm the relatively weaker resonance peaks, while the asymmetric peak shape might suggest several overlapping peaks.

2. Is the hole diameter d important for the PoC band structure? Please discuss the impact of the d/P ratio on the dispersion and far-field PoC response.

REPLY: We calculated the band structures of PoCs with varied diameters and periodicities. For convenience, we only considered the results along the Γ -X-M- Γ direction. As seen from Fig. S9, both the band frequency (dispersion) and field intensity of PoCs depend closely on their periodicities but are less relevant to diameters.

The d/P ratio also impacts the far-field absorption spectra of PoCs. However, compared with periodicity, diameter causes a relatively smaller frequency shift (Fig. S8a). This phenomenon can be understood through the normalized FFT amplitude maps shown in Fig. S8b. The strength of FFT amplitudes decreases with diameters but the reciprocal lattice vectors remain unchanged. The decreased FFT amplitude requires IFCs to shift towards the center, leading to a lower resonance frequency. But this effect is not as significant as that caused by the change of reciprocal lattice vectors.

ACTIONS: Figures S8 and S9 have been provided in revised Supplementary Information to show the impact of the d/P ratio on the far-field absorption and band structures of PoCs. The sentence “The resonance of PoCs can also be tuned by controlling the diameter of holes, as seen from calculated absorption spectra in Supplementary Fig. S8a. However, the diameter mainly relates to the amplitude intensity of FFT maps, imposing no influence on the reciprocal space points and thus leading to smaller frequency shifts compared to that of periodicity for the PoCs with a fixed d/P ratio (Supplementary Fig. S8b). Besides, we also calculated the band structures of PoCs with varied scales (Supplementary Fig. S9). The lattice periodicity also plays a major role in the band structure of PoCs. It can shift the band frequencies of PoCs and change relative field intensities. The diameter, however, has a limited impact on band structures.” has been added in the revised manuscript (Line 173).

Figure S8. **a**, Absorption spectra of PoCs with varied d/P ratios. **b**, IFCs and normalized FFT amplitude maps at the conditions marked by coloured dots in **a**.

Figure S9. Band structures of PoCs with varied diameters and periodicities. The map intensity was normalized. The gray curves represent light lines at certain periodicities.

3. In fig. 2c and S4a, IFCs do not cross the array momenta (the FFT map), yet the selected frequencies correspond to the absorption peaks. Please explain this mismatch. Also, the clear

absorption peak at around 925 cm^{-1} in fig. 2e should be explained.

REPLY: The absorption peaks are usually caused by the superposition of several resonance modes, which can be seen from the FFT maps in new Fig. 3c. Such effect requires the IFC contours to get close to several reciprocal space points. And sometimes, the IFC contours don't exactly cross those points.

ACTIONS: In the revised manuscript, the near-field images close to resonance frequencies have been shown in new Fig. 3a (please see the response to Comment 1.1). Figure 3c shows the IFCs and FFT maps, by which the mode order assignment can be made. According to Fig. 3c, the absorption peak at 925 cm^{-1} was assigned to the $(\pm 2, 0)$ order Bragg resonances.

4. In fig. 3, the near-field mapping frequency of 903.8 cm^{-1} does not correspond to any absorption feature of PoC (according to fig. 3g). Please show the near-field scans corresponding to the resonant modes in the structure.

REPLY: We have conducted near-field measurements at resonance frequencies of PoCs. Figure 3 shows the near-field amplitude images of the PoC with $P = 2.3\text{ }\mu\text{m}$ at three typical frequencies (please see the response to Comment 1.1). In Fig. 4 (previous Fig. 3), new results of the PoCs with $P = 1.8$ and $1.3\text{ }\mu\text{m}$ at resonance frequencies have been provided. Images in new Figs. 5 and 6 have also been updated.

ACTIONS: In the revised manuscript, near-field images and corresponding simulations at resonance frequencies have been updated in all the figures.

Figure 4. Tuning PoC modes and resonances by varying lattice periodicity. **a,b**, Near-field amplitude images of PoCs with different periodicities. **c-d**, Corresponding simulated electric field distributions (normalized). **e**, Calculated absorption coefficients of PoCs as a function of frequency and periodicity. White, blue, and red dashed curves represent the (0, 0), ($\pm 1, 0$), and (0, ± 1) order branches. **f**, Absorption spectra of PoCs. The yellow and green dots indicate the frequencies in **a** and **b**. **g,h**, Corresponding FFT maps of the experimental near-field images in **a** and **b**. The yellow and green curves represent the calculated IFCs of PhPs at 871 and 913 cm^{-1} , respectively.

5. Different PoC dispersion branches visible in fig. 3g and 3h should be explained.

REPLY: In previous Fig. 3g and 3h (new Figs. 4e,f), the absorption peak at 824 cm^{-1} corresponds to the intrinsic transverse optical (TO) phonon resonances of $\alpha\text{-MoO}_3$, which is independent of periodicities. Absorption peaks originated from polariton resonances shift with periodicities. Different-order modes yield several branches in new Fig. 4e and their intensities (contrasts) are determined by the matching condition between polariton wavevectors and lattice constants (please see more responses to Comment 5.1).

ACTIONS: The TO phonon frequency has been pointed out in new Figs. 4e,f, Figs. 5g,h, and Figs.6 c,e. A sentence “To have an in-depth analysis, we calculated far-field absorption map

(Fig. 4e) and corresponding spectra (Fig. 4f) and found that the polariton resonance peaks of our PoCs shift towards higher frequencies with decreasing P and form several dispersion branches, while the TO phonon frequency remains unchanged.” has been added in the revised manuscript. The different branches in Fig. 4e have been labelled.

5.1 Please explain why the absorption maximum for $P = 1.3 \mu\text{m}$ does not correspond to the zero-order resonance as for the other two period values.

REPLY: It can be seen from new Fig. 4e that for a given periodicity, the strength of (0, 0) order resonance is not always the highest one. We attribute this phenomenon to the optimal resonance condition, that is, IFCs intersect reciprocal points with high amplitude as many as possible. For PoCs with $P = 1.3 \mu\text{m}$, the superposition of (0, ± 1) order resonances is stronger than the single (0, 0) order resonance, leading to the absorption maximum at 871 cm^{-1} .

ACTIONS: The optical resonance condition has been highlighted in Line 167 “Besides peak shift, the absorption strength also varies with P and reaches the maximum at the optimal resonance condition, that is, IFCs intersecting the most reciprocal space points with high amplitude.” The field distribution and corresponding FFT images of the PoC with $P = 1.3 \mu\text{m}$ have been provided in Fig. S7a to have an in-depth understanding of the optimal resonance condition. As a comparison, the calculated results of the PoC with $P = 1.5 \mu\text{m}$ at 861 cm^{-1} (the condition that reaches the highest absorption in new Fig. 4e) have been given in Fig. S7b.

Figure S7. Simulated electric field distribution images of PoCs with $P = 1.3$ (a) and $1.5 \mu\text{m}$ (b) at resonance frequencies. c,d Corresponding FFT maps of the images in a and IFC contours (curves).

Minor problems

1. In line 87, what does the “resonance frequency” mean? PhCs generally have multiple bands at different frequencies. Also, please clarify the meaning of “frequency shift” – shift relative to what frequency?

REPLY: The frequency shift is referred to the certain resonance frequency of the square lattice with $P_x = P_y = P_0$. Here we consider the frequency with the highest resonant absorption (instead of phonon absorption) and study its dependence on the periodicities along the x - and y -directions. Please see the response to Minor Comment 3 for more details.

ACTIONS: A sentence “Using rectangular-type PoCs with a fixed defect geometry and size as an example, their resonance frequencies ($\omega_j^r, j = x, y$) are closely related to periodicity (P_j). Based on the Rayleigh-Wood anomaly,⁴⁰ the frequency shift of isotropic PoCs, denoted by $\Delta\omega_j^r = \omega_j^r - \omega_0^r$, where ω_0^r is the polariton resonance frequency of the square lattice with $P_x = P_y = P_0$, decreases reciprocally with P_j along both x and y directions (Fig. 1a, see supplementary Note 1 for details), resulting from the in-plane isotropic wavevectors (Fig. 1a,

inset).” has been added in Line 81.

2. I am not sure what is the importance of the section “Scaling laws of photonic crystals”. It seems disconnected from the following discussion. If this section is meant to introduce the concept of hyperbolic phonon-polaritons and hyperbolic PhC based on hyperbolic materials, then please provide references and clearly discuss these two concepts in consecutive manner.

REPLY: To make our discussion more concentrated and avoid possible misleading, the subtitle of “Scaling laws of photonic crystals” have been changed to “Scaling laws of polaritonic crystals”. We have also deleted the contents about photonic crystals in this section and just focused on polaritonic crystals. All conclusions in the main text are made within this regime.

2.1 Particularly, the discussion in lines 90-97 is very confusing. It seems that authors mixed up two very different topics – anisotropic polaritons and photonic crystals. For example, what is the meaning of $\epsilon_{x,y}$? Is this the permittivity of the polaritonic material? Please clarify the discussion.

REPLY: As we focus on polaritonic crystals, $\epsilon_{x,y}$ here is indeed the permittivities of polaritonic media at different crystalline axes.

3. How was the expression in line 88 for $\Delta\omega_j$ derived? This expression must be explained in details in supplementary information.

REPLY: The previous expression of $\Delta\omega_j$ was obtained by a combination of numerical simulation and plot fitting. The absorption spectra against different periodicities were obtained by full-wave simulations with proper material inputs. Considering Bloch modes in the system, we arbitrarily used an exponential decay equation $y = Ae^{-\frac{P_j}{P_0}S_j} + y_0$ to fit the

extracted $\Delta\omega_j$ and found a reasonably good agreement (Fig. R1).

Figure R1. Comparison between numerical data and two fitting results.

During revision, we have realized that the resonance frequency as well as frequency shift can be derived analytically through Rayleigh-Wood anomaly that is commonly used in plasmonic metamaterials [Photronics, 6, 75 (2019)]. At this condition,

$$\mathbf{k}_p = \mathbf{k}_{\parallel} + m\mathbf{G}_x + n\mathbf{G}_y$$

where $\mathbf{k}_p = \sqrt{\mathbf{k}_x^2 + \mathbf{k}_y^2}$ and \mathbf{k}_{\parallel} represent in-plane polariton wavevector and in-plane incident wavevector; $\mathbf{G}_x = \frac{2\pi}{P_x}\hat{x}$ and $\mathbf{G}_y = \frac{2\pi}{P_y}\hat{y}$ are reciprocal lattice vectors for the periods P_x and P_y ; m and n are diffraction orders. For normal incidence, $\mathbf{k}_{\parallel} = 0$, and the resonance frequency (ω_j^r) is solved to be $\omega_x^r = \frac{c}{\sqrt{\epsilon_x^{eff}}} \frac{m}{P_x}$ at the x direction and $\omega_y^r = \frac{c}{\sqrt{\epsilon_y^{eff}}} \frac{n}{P_y}$ at the y

direction. Here we calculate through the dispersion relation of surface polaritons $\mathbf{k}_p = \frac{2\pi\omega}{c}\sqrt{\epsilon_{eff}}$ where ϵ_{eff} is the effective permittivity. For volume-confined polaritons, they can also be treated as two-dimensional surfaces with effective conductivities. Finally, we can reach the equation for frequency shift ($\Delta\omega_j^r$)

$$\Delta\omega_x^r = \frac{mc}{\sqrt{\epsilon_x^{eff}}} \left(\frac{1}{P_x} - \frac{1}{P_0} \right) = \frac{mc}{\sqrt{\epsilon_x^{eff}}} \frac{P_0 - P_x}{P_x P_0}$$

for the x -packed arrays, and

$$\Delta\omega_y^r = \frac{nc}{\sqrt{\varepsilon_y^{eff}}} \left(\frac{1}{P_y} - \frac{1}{P_0} \right) = \frac{nc}{\sqrt{\varepsilon_y^{eff}}} \frac{P_0 - P_y}{P_y P_0}$$

for the y-packed arrays.

To double check the validity of the above equations, we used a simple function $y = A(1 - x)/x$ to fit the numerical results, as $\frac{mc}{P_0\sqrt{\varepsilon_x^{eff}}}$ and $\frac{nc}{P_0\sqrt{\varepsilon_y^{eff}}}$ are independent of P_x and P_y . A good agreement can be found in Fig. R1. We thus use this method to qualitatively describe the scaling law of polaritonic crystals with different permittivities.

ACTIONS: We have updated the section “Scaling laws of polaritonic crystals” (please see the response to Minor Comment 1). Besides, Supplementary Note 1 and Fig. S1 have been added in the revised Supplementary Information for more details about the derivation of scaling laws. Ref. 40 has been added in the revised main text.

Figure S1. **a**, Calculated absorption spectra of hyperbolic PoCs with a fixed P_y but varied P_x . **b**, Extracted frequency shifts and fitting results.

3.1 What is the importance of this expression? I don't see any use of it in the following analysis.

REPLY: The scaling law provides an overall picture of the influence of in-plane anisotropy on the resonance of polaritonic crystals. The remarkable difference among isotropic, elliptic, and hyperbolic polaritonic crystals not only highlights the novelty of our work, but implies

the key advantage of our hyperbolic polaritonic crystals, that is, robustness against periodicity variation at the polariton-forbidden direction.

3.2 Please show how the polaritonic dispersion relation is accounted for in the expression.

REPLY: We have used Rayleigh-Wood anomaly to qualitatively calculate frequency shifts in the revised manuscript (please see the response to Comment 3). The dispersion relation is used to connect resonance frequencies and reciprocal lattice vectors during calculation, which is not shown in the final expression.

3.3 Similar to the question 1 above, how does this expression correlate to the band structure of the PoC?

REPLY: In fact, this expression is just a qualitative description of the resonance frequency shifts against periodicity variations, which doesn't relate to the band structures of polaritonic crystals directly.

4. In line 150, please provide reference or analytical expression which supports the claim that quasi-flat band indicates the anisotropic Bloch mode.

REPLY: Thanks for identifying this loose expression. The quasi-flat band indicates the potential of our PoCs in self-collimation [*IEEE J. Sel. Top. Quantum Electron.*, **8**, 1246 (2002)], due to their in-plane hyperbolic IFCs.

ACTIONS: The sentence “The quasi-flat band around 846.9 cm⁻¹ further confirms the extreme anisotropy of the Bloch mode.” has been changed into “The quasi-flat around 846.9 cm⁻¹ demonstrates the self-collimation effect in the PoC, because of its hyperbolic IFCs.” Ref. 44 has been cited in the revised manuscript.

5. In lines 171-172 of the main text, fig. S4b is said to correspond to $P = 1.6 \text{ um}$, but in the SI

it is shown to correspond to 2.3 μm .

REPLY: Thanks for pointing out this mistake. We didn't fabricate samples with $P = 1.6 \mu\text{m}$.

It was a typo, which has been corrected in the revised manuscript.

ACTIONS: The periodicity of this PoC has been corrected to 2.3 μm in the main text.

6. Spectral absorption profile for 30 deg (green) in fig. 4h seems to disagree with the results in fig. 4g. The green dot in 4g supposed to be the peak position, but the actual peak is to the left from it.

REPLY: Thanks for this good point. The misalignment was caused by the slightly different scales between images and plots during figure panel arrangement. The alignment has been amended in the updated Fig. 5g (previous Fig. 4g).

ACTIONS: Figure 5g has been updated.

Figure 5. Low-symmetry Bloch modes in rotated hyperbolic PoCs. **a-c**, Resonant interference patterns of PoCs with the same periodicity ($P = 1.3 \mu\text{m}$) but different rotation angles. **d-f**, Normalized simulated electric field distributions. **g**, Calculated absorption coefficients of PoCs as a function of frequency and rotation angle. **h**, Absorption spectra of PoCs. The

coloured dots mark the frequencies in **a-c. i-k**, Corresponding FFT maps of the experimental interference patterns in **a-c**. The black, yellow, green, and purple curves represent the calculated IFCs of PhPs at 886, 879, and 892 cm^{-1} , respectively.

Reviewer #2

The authors fabricate a hyperbolic polaritonic crystal out of the in-plane anisotropic crystal MoO₃. Although the realization of a hyperbolic polaritonic crystal has been demonstrated in previous works (see Dickson, Wayne, et al. "Hyperbolic polaritonic crystals based on nanostructured nanorod metamaterials." *Advanced Materials* 27.39 (2015): 5974-5980) in this present work the authors focus on experimental imaging of highly asymmetric Bloch modes in the natural hyperbolic crystal MoO₃ and tune them by varying array periodicity and array orientation.

REPLY: We highly appreciate the reviewer's positive evaluation of our work. We studied both theoretically and experimentally the low-symmetry Bloch modes in polaritonic crystals rooted in in-plane hyperbolic phonon polaritons in α -MoO₃ and resultant tuning methods and extraordinary robustness. The work by D. Wayne, et al has been cited in the revised manuscript (Ref. 30). Notably, they studied hyperbolic polaritonic crystals composed of nanorod metamaterials with out-of-plane hyperbolicity but in-plane isotropy. Their PoCs are similar to those made from hBN (that is, isotropic PoCs) and are fundamentally different from our in-plane hyperbolic PoCs in our work.

1. In general, some of the main claims of the manuscript are not supported neither by experiment nor by theory:

-The authors claim that the Bloch modes are robust against defects and disorder without showing any experimental/theoretical proof. The authors name by "disorder" what is indeed a "lattice rearrangement".

-The authors claim that the Bragg resonances exhibit robust properties to lattice

rearrangement by showing a particular example of a lattice arrangement (diamond-like lattice shape). However, this statement does not hold true for other lattice arrangements (for example, if the lattice is rearranged such that the holes are displaced along the Y axis).

REPLY: Indeed, the Bragg resonances in our work are not robust to any types of lattice rearrangement. In our manuscript, we highlight the robust properties only along the forbidden direction of hyperbolic dispersion. This can be seen from the example of diamond-like lattice shape in Fig. 6 (previous Fig. 5). Actually, we extended such concept by introducing a set of rectangle lattices shown in Figs. 6c-e, and found that the periodicity variation in the forbidden direction (P_y) plays a minor role in the resonance frequency (absorption peak position) if P_x is fixed.

As mentioned by the reviewer, if the lattice is rearranged such that the holes are displaced along the y axis, then P_x is changed. This would no doubt affect the resonance frequency and is not the main claim of our work. To make our statement more precise, we emphasized the robustness against lattice rearrangement only subject to the fixed period in the x axis in our revised manuscript.

ACTIONS: In the revised manuscript, we have pointed out that the robustness of our hyperbolic PoCs is only valid for lattice rearrangement at certain directions in the polariton-forbidden area. Related contents in Abstract, Introduction and Conclusion parts has been updated.

2. The authors should provide more experimental data such as FTIR or nano-FTIR spectra of the arrays.

REPLY: As the reviewer suggested, we measured the FTIR spectrum of a PoC with $P = 2.3$

μm as an example. As shown in Fig. S4 below, the reflectance spectrum shows a strong peak at 817 cm^{-1} and a small peak at 1006 cm^{-1} , which should be attributed to the intrinsic TO and LO phonon resonance of $\alpha\text{-MoO}_3$, respectively. Unfortunately, this strong peak shows quite a large linewidth, making the resonance peaks (838 cm^{-1} , 892 cm^{-1} in new Fig. 4f) indistinguishable, although the asymmetric peak shape might suggest several overlapping peaks. We did not measure the FTIR spectra for other arrays due to their smaller areas (for example, only $6.5\times 6.5\ \mu\text{m}^2$ for $P = 1.3\ \mu\text{m}$). However, we can expect that if we have a larger array and non-absorptive substrate, the FTIR signal should evolve according to the calculated dispersion map. More systematic spectral responses of $\alpha\text{-MoO}_3$ arrays are definitely important and will be studied in the future. In the current manuscript, we stress and focus on the significant findings of configurable low-symmetry Bloch modes by near-field imaging.

ACTIONS: Figure S4 has been provided in the revised Supplementary Information.

Figure S4. Reflectance spectrum of the PoC composed of 5×5 hole arrays with $P = 2.3\ \mu\text{m}$. The signal was normalized to the unpatterned $\alpha\text{-MoO}_3$ region. The peaks at 817 and 1006 cm^{-1} indicate the TO and LO phonon resonance of $\alpha\text{-MoO}_3$, respectively, and overwhelm the relatively weaker resonance peaks, while the asymmetric peak shape might suggest several overlapping peaks.

3. Quality of data presentation is a bit poor.

REPLY: To improve the readability of our manuscript, we have provided a new figure (Fig. 3)

to show interference patterns at different resonance frequencies. Figure 4 has been re-arranged. Coloured dots have been added (Figs. 4e,f, Figs. 5g,h and Fig. 6c) to indicate resonance frequencies considered in experiments and simulations.

4. Some major comments to address are listed in the following:

Figure 3 and 4:

The authors should explain why simulated absorption spectra (panels g and h) do not match experiment at all. For instance, in Figure 3, for $P = 2.3 \mu\text{m}$ the (0,0) order resonance appears at 840 cm^{-1} and not at 903 cm^{-1} (big difference of about 60 cm^{-1}) as in the experimental image. In addition, why in panels e-f the simulated field distribution matches the experimental images while in the absorption spectra is completely off?

REPLY: Because of the narrow bandwidth of our CO_2 light source, in the previous manuscript, we just showed near-field results at specific frequencies that are far away from the resonance frequency. During revision, we have re-conducted near-field measurements using a QCL light source, which allows us to test at the frequencies close to resonances.

ACTIONS: In the revised manuscript (Figs. 3-6), all the near-field images have been updated, as well as simulations and IFCs.

In general, it is not clear how the mode order assignment is made. The authors should show mode field distributions for the different modes (0, 0), (± 1 , ± 1)... as well as the corresponding IFC contours and FFT map profiles (as in Supplementary Figure S2). In panels g, authors should relate mode orders to the mode branches.

REPLY: According to the reviewer's suggestion, we have performed FFT on near-field interference patterns at (or close to) resonance frequencies and compared with IFCs to show

the mode extraction. The results have been shown in new Fig. 3c.

ACTIONS: Figure 3 has been added to show the mode order assignment for PoCs with $P = 2.3 \mu\text{m}$. The branches corresponding to the $(0, 0)$, $(0, \pm 1)$, and $(\pm 1, 0)$ order resonances have been marked in Fig. 4e. Figure S7 has been added in the revised Supplementary Information to show the mode order assignment for PoCs with $P = 1.3$ and $1.5 \mu\text{m}$.

Figure 3. Frequency-dependent collective modes in hyperbolic PoCs. **a**, Near-field amplitude images of PoCs ($P = 2.3 \mu\text{m}$) at frequencies close to polariton resonance frequencies. **b**, Corresponding electric field distribution images (normalized). **c**, IFC contours (red curves) and FFT of the simulated images in **a**. The mode orders were determined by the contrast of FFT maps as well as the intersections between IFC contours and reciprocal space points (orange circles).

Figure 4. Tuning PoC modes and resonances by varying lattice periodicity. **a,b**, Near-field amplitude images of PoCs with different periodicities. **c-d**, Corresponding simulated electric field distributions (normalized). **e**, Calculated absorption coefficients of PoCs as a function of frequency and periodicity. White, blue, and red dashed curves represent the $(0, 0)$, $(\pm 1, 0)$, and $(0, \pm 1)$ order branches. **f**, Absorption spectra of PoCs. The yellow and green dots indicate the frequencies in **a** and **b**. **g,h**, Corresponding FFT maps of the experimental near-field images in **a** and **b**. The yellow and green curves represent the calculated IFCs of PhPs at 871 and 913 cm^{-1} , respectively.

Figure S7. Simulated electric field distribution images of PoCs with $P = 1.3$ (**a**) and 1.5 μm (**b**) at resonance frequencies. **c,d** Corresponding FFT maps of the images in **a** and IFC contours (curves).

The electric field shown in panels d, e and f is E or E_z ? Is this field normalized?

REPLY: The simulated electric fields are the absolute value of E ($|E|$), which have been normalized at each frequency.

ACTIONS: We have pointed out the normalized electric fields in the caption of Figs. 3–6.

Page 6: The authors write $P = 1.6 \mu\text{m}$ while Fig S4b shows $P = 2.3 \mu\text{m}$: “This resonance peak merges gradually with the (0, 0) resonance peak as P increases and reaches the maximum when $P = 1.6 \mu\text{m}$ at around 892.4 cm^{-1} (Fig. S4b).”

REPLY: Thank the reviewer for pointing out this mistake. We didn't fabricate samples with $P = 1.6 \mu\text{m}$. This was a typo in the main text which has been amended.

ACTIONS: The periodicity of this PoC has been corrected to $2.3 \mu\text{m}$ in the main text.

5. Some minor comments are listed below:

What is the fundamental difference of a 2D hole array of MoO_3 (with modes that only propagate in one direction) and a 1D hole array of h-BN (isotropic)?

REPLY: The fundamental difference lies in the distinct isofrequency contours (IFCs) of polaritons: hyperbolic for $\alpha\text{-MoO}_3$, and isotropic for h-BN. First, even for a 1D hole array of hBN, its modes still exist in two orthogonal directions (*Nat. Commun.*, 8, 15624 (2017)), thus leading to a resonance in the far field. Second, there is no rotation control for hBN PoCs. One can only obtain fixed Bloch modes and fixed Bragg resonances no matter what degree you rotate the 1D hole array of hBN. Third, the robustness against lattice rearrangement or periodicity variation in one direction cannot be realized in 1D array of hBN.

ACTIONS: A sentence “Such forbidden area makes $\Delta\omega_j^r$ robust against periodicity variation at certain direction (here, the y-direction). This unique property is hardly attainable in isotropic and elliptic PoCs and might be useful in disorder-tolerant optical resonators, directional light beaming, and other appealing applications.” has been added in Line 91 to

highlight the robustness against lattice rearrangement or period variation at the propagation-forbidden direction of our hyperbolic PoCs. A sentence “The natural in-plane hyperbolic PhPs enable rotation-tunable low-symmetry Bloch modes in our PoCs, offering a new degree of freedom for unparalleled resonance control, which is completely distinct from isotropic PoCs made of graphene or hBN.” has been added in Line 182 to emphasize the rotation tunability of our hyperbolic PoCs.

Can the authors show near-field images in other frequency ranges, i.e, showing the transition from hyperbolic to elliptic regimes?

REPLY: We have provided a near-field image and corresponding simulation at 987 cm^{-1} where $\alpha\text{-MoO}_3$ sustains in-plane elliptic polaritons. The obtained interference pattern differs from those highly directional ones within hyperbolic bands.

ACTIONS: Figure 6 has been provided in the revised Supplementary Information. The sentence “As a comparison, the near-field image at 987 cm^{-1} is provided in Supplementary Fig. S6c with a significantly different and reduced anisotropic interference pattern due to the in-plane elliptic dispersion of PhPs.” has been added in Line 156.

Figure S6. Near-field interference patterns and corresponding field distribution images of the PoC with $P = 2.3 \mu\text{m}$ at 904 cm^{-1} (a), $P = 1.8 \mu\text{m}$ at 904 cm^{-1} (b), the PoC with $P = 1.3 \mu\text{m}$ at 904 cm^{-1} (c) and 987 cm^{-1} (d).

Reviewer #3

In this manuscript, the authors experimentally explored a polaritonic crystal by patterning the alpha MoO₃ slabs. In the infrared region, the material shows hyperbolic phonon polaritons which are modified due to geometrical structuring such as changing the lattice constant or rotating the lattice of the pattern with respect to the crystal axes of alpha MoO₃. The authors show that the resonance shifts do not occur when the periodicity along the forbidden direction of the hyperbolic IFC is changed.

I have a general comment on the novelty and relevance of this work. Firstly, as stated in point 1) below, the claim of “robustness” of the polariton to disorder in one direction may be useless in practical situations. Secondly, regarding the novelty of the idea itself, it may be important to note that a theoretical proposal for the same system was published last year (this is reference 37 in the current manuscript):

Capote-Robayna, Nathaniel, Olga G. Matveeva, Valentyn S. Volkov, Pablo Alonso-González, and Alexey Y. Nikitin. "Twisted polaritonic crystals in thin van der Waals slabs." *Laser & Photonics Reviews* 16, no. 9 (2022): 2200428.

Hence the novelty of the idea itself is lessened. I leave to the editor the question of whether in the light of these considerations, this manuscript is worthy of publication in *Nature Communications*.

REPLY: Ref. 37 theoretically studied the rotation-tuned spectral responses of polaritonic crystals made from α -MoO₃ very recently. However, little attention was paid to low-symmetry Bloch modes and their near-field responses in polaritonic crystals, which arises from the unparalleled features of in-plane hyperbolic polaritons hosted by α -MoO₃. In

our manuscript, we systematically correlate the Bloch modes (near-field), Bragg resonances (far-field), and FFTs (k -space) to gain a deep insight into the hyperbolic PoC.

Besides, such low-symmetry modes endow our polaritonic crystals robustness against lattice rearrangement at the forbidden direction, bearing great potentials in disorder-tolerant optical resonators, directional light beaming, and other appealing applications. This extraordinary effect is also not investigated and not achievable based on the PoC design in Ref. 37 nor other papers in the field.

Moreover, according to the reviewers' critical comments and advice, we have substantially amended both the theoretical and experimental data/analysis (including improvement of most of the main and supporting figures and corresponding discussions), and improved the scientific rigor of the work. We hope that our revised manuscript can now meet the high standard of *Nature Communications*.

Additionally, I have a few technical comments which are listed below:

1) One of the strongly pitched novelty is the authors' claim of robustness (eg. "disorder tolerant optical resonator" in line 99) -- which seems very far fetched. In reality, their system only is robust to periodicity variation in one direction. Typically disorder would occur along all directions. Could the authors comment on what kind of experimental scenarios would give rise to this very specific kind of "disorder"?

REPLY: Indeed, one of the novelties of our work is the robustness of our hyperbolic PoCs against lattice rearrangement, that is, periodicity variation at the forbidden direction of hyperbolic dispersion as the reviewer mentioned. The disorder in our work only refers to the periodicity variation in one direction, which is different from conventional disorder systems

along all directions. Thus, such disorder does not resemble experimental defects, but can be used for specific needs and designs. For example, we can easily modulate polariton resonances in the elliptic band of α -MoO₃ by changing the period in the forbidden direction, while maintaining a fixed resonance in the hyperbolic band. This cannot be achieved using conventional photonic crystals or isotropic polaritonic materials.

ACTIONS: To avoid overclaim, in the revised manuscript, we have amended all the statements regarding defects or disorder and just focus on the robustness against lattice rearrangement at the propagation-forbidden direction.

2) A reference needs to be added for the exponential dependence of the frequency shift on the periodicity in line 88.

REPLY: In the revised manuscript, we qualitatively describe the dependence of frequency shifts on the periodicities based on the Rayleigh–Wood anomaly [*Photonics*, 6, 75 (2019)] and confirm the validity of this method by comparing it to numerical results.

At this condition,

$$\mathbf{k}_p = \mathbf{k}_{\parallel} + m\mathbf{G}_x + n\mathbf{G}_y$$

where $\mathbf{k}_p = \sqrt{\mathbf{k}_x^2 + \mathbf{k}_y^2}$ and \mathbf{k}_{\parallel} represent in-plane polariton wavevector and in-plane incident wavevector; $\mathbf{G}_x = \frac{2\pi}{P_x}\hat{x}$ and $\mathbf{G}_y = \frac{2\pi}{P_y}\hat{y}$ are reciprocal lattice vectors for the periods

P_x and P_y ; m and n are diffraction orders. For normal incidence, $\mathbf{k}_{\parallel} = 0$, and the resonance

frequency (ω_j^r) is solved to be $\omega_x^r = \frac{c}{\sqrt{\epsilon_x^{eff}}} \frac{m}{P_x}$ at the x direction and $\omega_y^r = \frac{c}{\sqrt{\epsilon_y^{eff}}} \frac{n}{P_y}$ at the y

direction. Here we calculate through the dispersion relation of surface polaritons $\mathbf{k}_p =$

$\frac{2\pi\omega}{c}\sqrt{\epsilon_{eff}}$ where ϵ_{eff} is the effective permittivity. For volume-confined polaritons, they can

also be treated as two-dimensional surfaces with effective conductivities. Finally, we can

reach the equation for frequency shift ($\Delta\omega_j^r$)

$$\Delta\omega_x^r = \frac{mc}{\sqrt{\varepsilon_x^{eff}}} \left(\frac{1}{P_x} - \frac{1}{P_0} \right) = \frac{mc}{\sqrt{\varepsilon_x^{eff}}} \frac{P_0 - P_x}{P_x P_0}$$

for the x -packed arrays, and

$$\Delta\omega_y^r = \frac{nc}{\sqrt{\varepsilon_y^{eff}}} \left(\frac{1}{P_y} - \frac{1}{P_0} \right) = \frac{nc}{\sqrt{\varepsilon_y^{eff}}} \frac{P_0 - P_y}{P_y P_0}$$

for the y -packed arrays.

To double check the validity of the above equations, we used a simple function $y = A(1 - x)/x$ to fit the numerical results (Fig. S1a), as $\frac{mc}{P_0\sqrt{\varepsilon_x^{eff}}}$ and $\frac{nc}{P_0\sqrt{\varepsilon_y^{eff}}}$ are independent of P_x and P_y . A good agreement can be found in Fig. S1b. We thus use this method to qualitatively describe the scaling law of polaritonic crystals with different permittivities.

ACTIONS: We have updated the section “Scaling laws of polaritonic crystals”. A sentence “Using rectangular-type PoCs with a fixed defect geometry and size as an example, their resonance frequencies (ω_j^r , $j = x, y$) are closely related to periodicity (P_j). Based on the Rayleigh-Wood anomaly,⁴⁰ the frequency shift of isotropic PoCs, denoted by $\Delta\omega_j^r = \omega_j^r - \omega_0^r$, where ω_0^r is the polariton resonance frequency of the square lattice with $P_x = P_y = P_0$, decreases reciprocally with P_j along both x and y directions (Fig. 1a, see supplementary Note 1 for details), resulting from the in-plane isotropic wavevectors (Fig. 1a, inset)” has been added in Line 81. Besides, Supplementary Note 1 and Fig. S1 have been added in the revised Supplementary Information for more details about the derivation of scaling laws. Ref. 40 has been added in the revised main text.

Figure S1. **a**, Calculated absorption spectra of hyperbolic PoCs with a fixed P_y but varied P_x , from which the frequency shift of PoCs were obtained. **b**, Extracted frequency shifts and fitting results.

3) Figure 3 and its explanation in lines 163--169 is confusing. Figure 3a-c are all at a frequency of 903.8 cm⁻¹ according to the figure caption. Figure 3i-k are also presumably at the same frequency, since these are just spatial FFTs of Fig. 3a-c. However when the authors are explaining redshift of the (0,0) peaks in the paragraph starting at line 163, say "We attribute this phenomenon to the smaller reciprocal lattice vectors of PoCs with larger P, which require the shift of IFCs to satisfy the request of Bragg resonances, namely, intersections between IFCs and FFT harmonics. This tendency can be verified by the stronger contrasts of FFT harmonics that interact with IFCs in Figs. 3i-k". So the "FFT contrasts" that the authors use to explain the results are all for the same frequency of 903.8 cm⁻¹ and not at the location of the respective peaks.

REPLY: Because of the narrow bandwidth of our CO₂ light source, in the previous manuscript, we just showed near-field results at specific frequencies that are far away from the resonance frequency. During revision, we have conducted near-field measurements at (or close to) resonance frequencies of each PoCs and updated the near-field interference patterns

and corresponding numerical and analytical results in Figs. 3–6.

ACTIONS: In the revised manuscript (Figs. 3-6), all the near-field images have been updated, as well as simulations and IFCs. The resonance frequencies at which the near-field measurements were conducted have been pointed out by coloured dots in each plots.

Figure 3. Frequency-dependent collective modes in hyperbolic PoCs. **a**, Near-field amplitude images of PoCs ($P = 2.3 \mu\text{m}$) at frequencies close to polariton resonance frequencies. **b**, Corresponding electric field distribution images (normalized). **c**, IFC contours (red curves) and FFT of the simulated images in **a**. The mode orders were determined by the contrast of FFT maps as well as the intersections between IFC contours and reciprocal space points (orange circles).

Figure 4. Tuning PoC modes and resonances by varying lattice periodicity. **a,b**, Near-field amplitude images of PoCs with different periodicities. **c-d**, Corresponding simulated electric field distributions (normalized). **e**, Calculated absorption coefficients of PoCs as a function of frequency and periodicity. White, blue, and red dashed curves represent the $(0, 0)$, $(\pm 1, 0)$, and $(0, \pm 1)$ order branches. **f**, Absorption spectra of PoCs. The yellow and green dots indicate the frequencies in **a** and **b**. **g,h**, Corresponding FFT maps of the experimental near-field images in **a** and **b**. The yellow and green curves represent the calculated IFCs of PhPs at 871 and 913 cm^{-1} , respectively.

Figure 5. Low-symmetry Bloch modes in rotated hyperbolic PoCs. **a-c**, Resonant interference patterns of PoCs with the same periodicity ($P = 1.3 \mu\text{m}$) but different rotation angles. **d-f**, Normalized simulated electric field distributions. **g**, Calculated absorption coefficients of PoCs as a function of frequency and rotation angle. **h**, Absorption spectra of PoCs. The coloured dots mark the frequencies in **a-c**. **i-k**, Corresponding FFT maps of the experimental interference patterns in **a-c**. The black, yellow, green, and purple curves represent the calculated IFCs of PhPs at 886, 879, and 892 cm^{-1} , respectively.

Figure 6. Robust resonant modes against lattice rearrangement in hyperbolic PoCs. **a**, Near-field amplitude image of a diamond-lattice PoC with $P = 2.3 \mu\text{m}$ at 892cm^{-1} . **b**, Corresponding electric field distribution (normalized). **c**, Absorption spectra of PoCs with different lattice arrangements. The green dot represents the frequency in **a**. **d**, IFCs of PhPs at 839cm^{-1} . The color plots represent the normalized FFT amplitude maps of PoCs with diamond (left panel) and rectangle (right panel) lattices. The gray shaded areas indicate the forbidden areas of hyperbolic dispersion. **e**, Calculated absorption coefficients of PoCs as a function of frequency and length-width ratio (P_y/P_x). Inset is the schematic of the rectangular-type PoC. The purple dashed line at $P_y/P_x = 0.57$ marks the absorption curve (purple) shown in **c**. The black and cyan dashed lines indicate the frequencies of the TO phonon resonance and $(0, 0)$ polariton resonance, respectively.

4) The authors say in line 168, “For the PoC with $P = 1.3 \mu\text{m}$, the highest absorption peak is located at 871.3cm^{-1} , which should be attributed to the $(\pm 1, \pm 2)$ resonances (Supplementary Fig. S4a). This resonance peak merges gradually with the $(0, 0)$ resonance peak as P increases and reaches the maximum when $P = 1.6 \mu\text{m}$ at around 892.4cm^{-1} (Fig. S4b)”. However, in supplementary Figure S4b, the periodicity is chosen as $2.3 \mu\text{m}$ (according to Fig S4 caption) instead of the above stated 1.6 microns.

REPLY: Thank the reviewer for pointing out this mistake. We didn't fabricate samples with $P = 1.6 \mu\text{m}$. This was a typo in the main text which has been amended.

ACTIONS: The periodicity of this PoC has been corrected to $2.3 \mu\text{m}$ in the main text.

5) In line 193, the authors say: "Using the case with $\theta = 45^\circ$ as an example (Fig. 4k), the highest absorption peak at 844.3 cm^{-1} is the result of the combination of $(0, 0)$, $(\pm 1, 0)$, and $(0, \pm 1)$ order resonances, which can be derived from the relatively stronger contrast of corresponding FFT harmonics". Once again, this is not clear from Figure 4.

REPLY: New experiments and simulations have been conducted at resonance frequencies of each PoCs according to calculated absorption spectra. Relevant results have been provided in the revised manuscript (Figs. 3–6). Please see the response to Comment 3. All the analysis and discussions are made at the resonance conditions clarified by absorption spectra. The mode assignment of rotated PoCs is relatively complicated than that of PoCs with varied periodicities, because the oblique reciprocal space points in the former.

ACTIONS: The near-field images and relevant numerical and analytical results have been updated in Fig. 5 (previous Fig. 4). The sentences "We plot the IFC curves and rotated FFT distributions in Figs. 4i-k at the frequencies marked by vertical dashed lines in Fig. 4h where the absorption is highest. One can see that the intensity of absorption is strongly dependent on the interaction between IFCs and FFT harmonics. Similar to the phenomena observed from PoCs with varied P , the absorption reaches the maximum at the frequency where several different-order resonances superpose. Using the case with $\theta = 45^\circ$ as an example (Fig. 4k), the highest absorption peak at 844.3 cm^{-1} is the result of the combination of $(0, 0)$, $(\pm 1, 0)$, and $(0, \pm 1)$ order resonances, which can be derived from the relatively stronger contrast of corresponding FFT harmonics." have been re-written into "We plot the IFC curves in Figs. 5i-k at the resonance frequencies marked by coloured dots in Fig. 5h. The rotation of arrays changes reciprocal lattice vectors (FFT maps in Figs. 5i-k), while IFCs remain unchanged,

leading to shifted resonance frequencies similar to those of PoCs with different periodicities. However, the mode assignment and superposition of resonances are relative complicated in rotated PoCs, because of their oblique reciprocal lattice vectors.”

REVIEWERS' COMMENTS

Reviewer #1 (Remarks to the Author):

I appreciate the authors' effort to address the numerous comments given by reviewers. In general, I am satisfied with the quality of the revised manuscript. I have a few additional questions and comments that should be addressed before manuscript can be accepted for publication.

1.1 Regarding the robustness of hyperbolic PoC to lattice re-arrangement, since this novelty claim is emphasized several times throughout the manuscript, I suggest providing some more analysis on dependence of robustness to the displacement direction relative to the MoO₃ crystal axes and array orientation (similar to the case discussed in Fig. 5). For example, what happens when the crystal is tilted relative to the crystal axes? Is it still robust to any displacement?

1.2 The polaritonic resonance in the second Reststrahlen band is bound to the x-direction when the array is aligned to the crystal axis (Fig. 6a). Does it mean that the considered PoC is of 1D nature instead of 2D in this case? If so, then the claimed robustness can be easily explained as a consequence of this.

2.1 In the discussion of Fig. 2 (ln. 137-138), please indicate how exactly do you calculate the band structure.

2.2 Please show the partially opened bandgap in Fig. 2h. Also, please indicate the meaning of the color map in the same figure.

3. Please carefully proofread the text for grammatical errors and poor wording. Here are a few examples I noticed:

Ln. 35: "...artificial structures [that] can mold..."

Ln. 36: please consider re-writing the whole sentence. To what "material excitations" light can "strongly couple" in PhC? Polaritons exist (or not) irrespectively of PhC structure. And PoC is not formed merely because polaritons can be excited in a material the PhC is made of.

Ln. 51-52: please give a proper definition of polaritons. For example: a quasiparticle formed via coupling of light to collective oscillations of charges in a polaritonic material. Or something like that.

Ln. 61: what is enigmatic about Bloch modes?

Ln. 81: "...efficient light modulation..." – but PoC does not modulate light in a conventional meaning of the term. Please consider using alternative terminology.

Ln. 89: What do you mean by "different decay ratios along the x and y"? I don't see any discussion of losses.

Ln. 170: Should be Fig. 4f instead of 4g.

Ln. 204: "...is observed at the center of two holes" – I guess, it should be "between the holes".

Reviewer #2 (Remarks to the Author):

The authors have successfully addressed all my concerns and changed the manuscript substantially. I thus I recommend it for publication in Nature Communications.

Some minor comments from the revised manuscript are addressed in the following:

- In Figure 4 dashed line marking TO phonon peak is missing.
- In Figure Caption 4 there is a typo in the sentence "White, blue, and red dashed curves represent the (0, 0), (± 1 , 0), and (0, ± 1) order branches"; it should be written: "White, blue, and red dashed curves represent the (0, 0), (0, ± 1) and (± 1 , 0) order branches, respectively."
- In Figure Caption 5 there is a typo in the sentence: "The black, yellow, green, and purple curves represent the calculated IFCs of PhPs at 886, 879, and 892 cm⁻¹, respectively." It should be written: "Yellow, green, and purple ...".
- In Figure 5 the simulated absorption spectra for $\theta = 0^\circ$ can be removed from panel h as it is not supported by experimental data.

Reviewer #3 (Remarks to the Author):

The authors have addressed my concerns about the first version of the manuscript. The revised manuscript is fit to be published in the journal.

Manuscript ID: NCOMMS-22-49996

Manuscript title: Hyperbolic polaritonic crystals with configurable low-symmetry Bloch modes

Point-by-point responses to Reviewers' Comments

We are very grateful for all the comments from the editor and all the reviewers. These comments greatly improve the quality and readability of this paper. Revisions and responses to address your comments are presented below.

Reviewer #1

I appreciate the authors' effort to address the numerous comments given by reviewers. In general, I am satisfied with the quality of the revised manuscript. I have a few additional questions and comments that should be addressed before manuscript can be accepted for publication.

REPLY: We appreciate the reviewer's comments and suggestions, which greatly improved the quality of this manuscript to meet the high standards of *Nature Communications*.

Comment 1.1: Regarding the robustness of hyperbolic PoC to lattice re-arrangement, since this novelty claim is emphasized several times throughout the manuscript, I suggest providing some more analysis on dependence of robustness to the displacement direction relative to the MoO₃ crystal axes and array orientation (similar to the case discussed in Fig. 5). For example, what happens when the crystal is tilted relative to the crystal axes? Is it still robust to any displacement?

REPLY: We thank the reviewer for raising this good question. The calculated absorption coefficients of the PoCs tilted by 10°, 20°, 30°, and 40° as a function of frequency and

length-width ratio have been provided in Supplementary Figure 10 in the revised Supplementary Information. One can find that the resonance frequencies shift with the change of periods ($P_{y'}$) in the y' arrangement direction (off-normal to the x crystalline axis). This phenomenon differs from that for the un-rotated PoC in Fig. 6f and indicates that the relative position between crystalline orientation and arrangement direction, that is, θ , plays an essential role in the robustness of hyperbolic PoCs to lattice re-arrangement. Because the Bloch mode is mainly concentrated in the x direction, such robustness emerges when the lattice arrangement direction is normal to the x crystalline axis ($\theta = 0^\circ$), and the robustness vanishes with the deviation from the x crystalline axis, namely, the increase of θ .

ACTIONS: Supplementary Figure 10 has been added in the revised supplementary information. A sentence “The robustness emerges when the lattice arrangement direction is normal to the quasi-1D field distributions, namely, the x crystalline axis ($\theta = 0^\circ$), and the robustness gradually vanishes in tilted PoCs with increasing θ , as confirmed by the simulated absorption coefficient maps of rotated PoCs with varied periodicities in Supplementary Figure 10.” has been added in Line 220 to discuss the robustness in tilted PoCs.

Supplementary Figure 10. Calculated absorption coefficients of the PoCs tilted by different angles as a function of frequency and length-width ratio.

Comment 1.2: The polaritonic resonance in the second Reststrahlen band is bound to the x -direction when the array is aligned to the crystal axis (Fig. 6a). Does it mean that the considered PoC is of 1D nature instead of 2D in this case? If so, then the claimed robustness can be easily explained as a consequence of this.

REPLY: Indeed, in the second Reststrahlen band, the distribution of the electric field in our hyperbolic polaritonic crystals is highly directional and mainly concentrated in the x -direction. Thanks to the low-symmetry Bloch modes, our 2D-arranged polaritonic crystals can hold quasi-1D field distributions, which is hardly attainable by isotropic or elliptic polaritonic

crystals or conventional photonic crystals.

ACTIONS: A sentence “Note that the polariton resonance is intrinsically bound to the x crystallin axis in the considered frequency ranges, yielding quasi-1D field distributions in our 2D-arranged PoCs.” has been added in Line 218 to highlight the quasi-1D field distribution in our PoCs.

Comment 2.1: In the discussion of Fig. 2 (ln. 137-138), please indicate how exactly do you calculate the band structure.

REPLY: The calculation process of band structures has been clarified in the Methods section. “The band structure of PoCs was calculated numerically by the finite difference time domain (FDTD) method using Ansys Lumerical. In the simulation, a unit cell was placed on a SiO₂ substrate. Bloch boundary conditions were set up at the sidewalls of the unit cell, whereas the top and bottom surfaces were set as perfectly matched layers. Randomly polarized electric dipoles were used as the light source to excite different Bloch modes^{31,34}. The signals were recorded by randomly placed monitors. Because of the relatively high losses in the system, the simulation time should be long enough.”

Comment 2.2: Please show the partially opened bandgap in Fig. 2h. Also, please indicate the meaning of the color map in the same figure.

REPLY: The partially opened bandgap has been indicated by black arrows in Fig. 2h. The meaning of the colour map, that is, the electric field intensity ($|E|$), has also been indicated by the colour bar.

Comment 3: Please carefully proofread the text for grammatical errors and poor wording.

REPLY: We thank the reviewer for pointing out our inappropriate presentations. We have

amended accordingly and polished the manuscript carefully.

Here are a few examples I noticed:

- 1) Ln. 35: "...artificial structures [that] can mold..."

The sentence has been updated accordingly.

- 2) Ln. 36: please consider re-writing the whole sentence. To what "material excitations" light can "strongly couple" in PhC? Polaritons exist (or not) irrespectively of PhC structure. And PoC is not formed merely because polaritons can be excited in a material the PhC is made of.

The sentence has been updated to "Polaritonic crystals (PoCs) made from polaritonic media offer a promising route to controlling nano-light at the subwavelength scale."

- 3) Ln. 51-52: please give a proper definition of polaritons. For example: a quasiparticle formed via coupling of light to collective oscillations of charges in a polaritonic material. Or something like that.

A clear definition of polaritons has been provided via the sentence "Polaritons, hybrid quasiparticles originating from the coupling of photons and material excitations, open up a promising pathway..."

- 4) Ln. 61: what is enigmatic about Bloch modes?

The word "enigmatic" has been replaced by "Many".

- 5) Ln. 81: "...efficient light modulation..." – but PoC does not modulate light in a conventional meaning of the term. Please consider using alternative terminology.

The sentence has been updated to "One of the key virtues of PoCs is efficient light field control by adjusting structural configurations."

- 6) Ln. 89: What do you mean by “different decay ratios along the x and y”? I don’t see any discussion of losses.

Here, “decay ratios” represent different slopes of curves, instead of losses. The sentence has been changed into “As shown in Fig. 1b, in elliptic PoCs, $\Delta\omega_j^r$ also decreases with P_j but exhibits different descending tendencies along the x and y directions.” to avoid misleading.

- 7) Ln. 170: Should be Fig. 4f instead of 4g.

The mistake has been corrected in the revised manuscript.

- 8) Ln. 204: “...is observed at the center of two holes” – I guess, it should be “between the holes”.

The sentence has been updated to “The near-field image at 892 cm^{-1} is displayed in Fig. 6a, where the destructive interference is observed between the holes...”

Reviewer #2

The authors have successfully addressed all my concerns and changed the manuscript substantially. I thus I recommend it for publication in Nature Communications.

REPLY: We highly appreciate the reviewer’s positive evaluation and recommendation of this work.

Some minor comments from the revised manuscript are addressed in the following:

REPLY: We thank the reviewer for the good comments. We have amended carefully in the revised manuscript.

- 1) In Figure 4 dashed line marking TO phonon peak is missing.

The vertical dashed line indicating the TO phonon peak has been added in Fig. 4.

- 2) In Figure Caption 4 there is a typo in the sentence “White, blue, and red dashed curves represent the (0, 0), (± 1 , 0), and (0, ± 1) order branches”; it should be written: “White, blue, and red dashed curves represent the (0, 0), (0, ± 1) and (± 1 , 0) order branches, respectively.

The caption of Fig. 4 has been updated accordingly.

- 3) In Figure Caption 5 there is a typo in the sentence: “The black, yellow, green, and purple curves represent the calculated IFCs of PhPs at 886, 879, and 892 cm^{-1} , respectively.” It should be written: “Yellow, green, and purple ...”.

The caption of Fig. 5 has been amended.

- 4) In Figure 5 the simulated absorption spectra for $\theta = 0^\circ$ can be removed from panel h as it is not supported by experimental data.

The simulation result for $\theta = 0^\circ$ (gray curve) has been removed from Fig. 5h.

Other revisions made in the revised manuscript:

1. The data in Figs. 5c,f,k were measured or calculated at the frequency of 892 cm^{-1} which is not the resonance frequency of the PoC tilted by 45° . The correct frequency should be 905 cm^{-1} , as seen from the purple dot in Fig. 5h. We have replaced Figs. 5c,f,k by the results at 904 cm^{-1} (close to the resonance frequency) in the revised manuscript. This revision does not weaken our main conclusions.
2. The schematic in the inset of Fig. 6f (previous Fig. 6e) has been redrawn in the revised manuscript to show the unit cell of PoCs.